# Tmprss2 maintains epithelial barrier integrity and transepithelial sodium transport

Olivia J Rickman*, Emma Guignard, Thomas Chabanon, Giovanni Bertoldi, Muriel Auberson, Edith Hummler*

The mouse cortical collecting duct cell line presents a tight epithelium with regulated ion and water transport. The epithelial sodium channel (ENaC) is localized in the apical membrane and constitutes the rate-limiting step for sodium entry, thereby enabling transepithelial transport of sodium ions. The membrane-bound serine protease *Tmprss2* is co-expressed with the alpha subunit of ENaC. *αENaC* gene expression followed the *Tmprss2* expression, and the absence of Tmprss2 resulted not only in down-regulation of *αENaC* gene and protein expression but also in abolished transepithelial sodium transport. In addition, RNA-sequencing analyses unveiled drastic down-regulation of the membrane-bound protease CAP3/St14, the epithelial adhesion molecule EpCAM, and the tight junction proteins claudin-7 and claudin-3 as also confirmed by immunohistochemistry. In summary, our data clearly demonstrate a dual role of Tmprss2 in maintaining not only ENaC-mediated transepithelial but also EpCAM/claudin-7–mediated paracellular barrier; the tight epithelium of the mouse renal mCCD cells becomes leaky. Our working model proposes that Tmprss2 acts via CAP3/St14 on EpCAM/claudin-7 tight junction complexes and through regulating transcription of *αENaC* on ENaC-mediated sodium transport.

## Introduction

Transepithelial sodium transport and epithelial barrier integrity are closely linked and are implicated in various physiological processes. Transepithelial sodium transport is crucial for the regulation of fluid balance, electrolyte homeostasis, and blood pressure control; it primarily occurs through specialized transport proteins on the membranes of epithelial cells (Chanez-Paredes et al, 2021). The non–voltage-gated highly amiloride-sensitive epithelial sodium channel (ENaC) is encoded by the three subunit genes alpha (*Scnn1a*), beta (*Scnn1b*), and gamma (*Scnn1g*). ENaC is expressed in the tight epithelia of various tissues such as the kidney, colon, lung, and sweat glands; in the kidney, it is localized to the distal convoluted tubule, the connecting tubule, and the cortical collecting duct (Rotin & Staub, 2021). ENaC plays a major role in the regulation of vectorial transcellular sodium transport in the distal part of the nephron, allowing the passive movement of sodium from the tubular lumen towards the blood, and is largely responsible for the fine-tuning of sodium homeostasis (Ehret & Hummler, 2022). Many membrane-bound proteases have been identified as ENaC-activating proteases, also named CAPs (channel-activating proteases) (Anand et al, 2022). In the *Xenopus* oocyte expression system, the sequential proteolytic activation of ENaC intracellularly by Furin and at the cell surface by several proteases was proposed (Kleyman & Eaton, 2020).

Amongst those proteases is the widely expressed androgen-regulated type II transmembrane serine protease Tmprss2 (Vaarala et al, 2001), but its physiological role is still largely unknown. Tmprss2 is primarily known for its role in the proteolytic cleavage of proteins like the angiotensin-converting enzyme 2 (ACE2) (Wettstein et al, 2022) or the protease-activated receptor 2 (PAR2) involved in various signalling pathways (Wilson et al, 2005). Another endogenous substrate identified is the epithelial sodium channel (Donaldson et al, 2002; Sure et al, 2022). Tmprss2 has been shown to be inhibited by hepatocyte-activating inhibitor-2 (HAI-2) (Faller et al, 2014) and reported to activate the transmembrane serine protease matriptase (St14) to promote extracellular matrix degradation, prostate cancer cell invasion, tumour growth, and metastasis via overactivation of MET, the specific receptor of hepatocyte growth factor (Ko et al, 2015; Mukai et al, 2020). *Xenopus* oocyte expression studies co-expressing Tmprss2 and ENaC subunits revealed contradictory results; Donaldson and co-workers found a decrease in ENaC current and protein levels, which is not prevented by the addition of aprotinin (Donaldson et al, 2002), whereas others reported that Tmprss2 activates ENaC through the proteolytic cleavage of gamma ENaC (Faller et al, 2014; Sure et al, 2022).

Tmprss2 regulates ENaC function in *Xenopus* oocytes; however, its regulation of the epithelial cell adhesion molecule (EpCAM) and

---

Department of Biomedical Sciences, Faculty of Biology and Medicine, University of Lausanne, Lausanne, Switzerland

Correspondence: Edith.Hummler@unil.ch
Emma Guignard and Thomas Chabanon's present address is Haute Ecole d'Ingénierie, Sion, Switzerland
Giovanni Bertoldi's present address is Department of Medicine (DIMED), University of Padova, Italy
*Olivia J Rickman and Edith Hummler contributed equally to this work

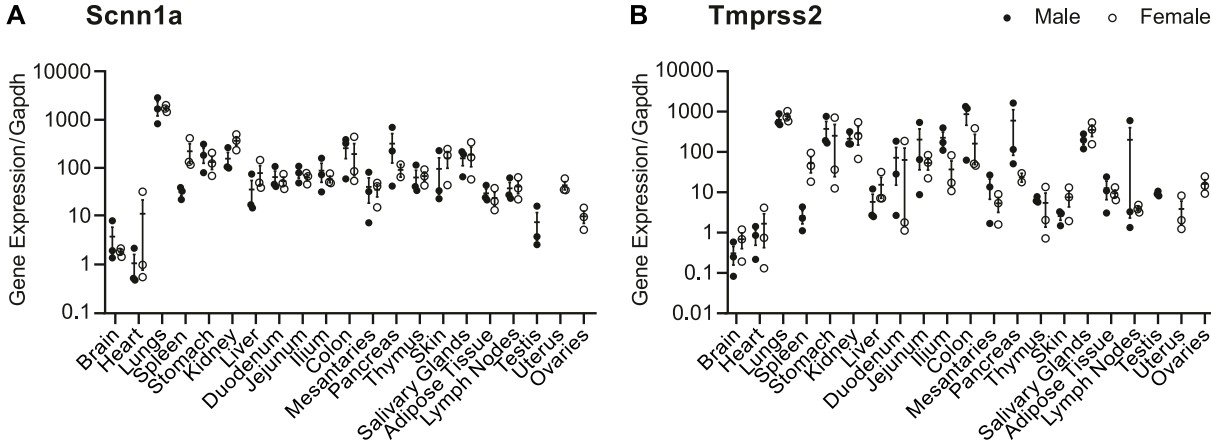

**Figure 1. *Scnn1a* and *Tmprss2* mRNA transcript levels are similar in various organs.**
**(A, B)** qRT–PCR expression levels of *Scnn1a* (A) and *Tmprss2* (B) in various organs from three male and three female C57BL/6J mice kept on a standard diet and normalized to *Gapdh*.

tight junction proteins is yet to be reported. EpCAM (also known as CD326) is a transmembrane glycoprotein involved in cell adhesion, signalling, migration, proliferation, and differentiation (Mohtar et al, 2020). Intramembrane proteolysis activates EpCAM, which involves shedding of its ectodomain and nuclear translocation of its intracellular domain conferring nuclear signalling and proliferation capacities (Maetzel et al, 2009). As a surface marker, it is found in the lateral membrane including the tight junction of the intestinal epithelium (Wu et al, 2013), and affects the progression, treatment, and diagnosis of many adenocarcinomas (Went et al, 2004). Knockout of EpCAM in murine models down-regulated the claudin-7 protein abundance levels in the intestine (Lei et al, 2012), and loss of EpCAM signalling leads to abnormal development of intestinal epithelial cells causing congenital tufting enteropathy (Guerra et al, 2012).

Both the overexpression and knockout of the membrane-bound serine protease Prss8 in mouse skin led to impaired barrier function because of increased tight junction permeability, altered claudin-1 and zonula occludens-1 (ZO-1) expression, and loss of occludin (Leyvraz et al, 2005; Frateschi et al, 2011; Crisante et al, 2014); a similar phenotype was also observed in matriptase knockout mice (List et al, 2002). Only recently, Higashi and co-workers (2023) presented a working model of tight junction maintenance by EpCAM and membrane-associated serine proteases (MASPs) in MDCK cells (Higashi et al, 2023). They show that the cleavage of EpCAM by MASPs induces the release of claudin-7 from the EpCAM/claudin-7 complex to maintain and repair the tight junction barrier. Upon knockout of selected MASPs, the barrier function is compromised with cells exhibiting increased size and leaky tight junctions (Higashi et al, 2023).

In this study, we studied Tmprss2 in the mouse kidney, manipulated the mouse cortical collecting duct (mCCD$_{cl1}$) cell line by CRISPR/Cas9 gene editing, and treated the cells with aldosterone and amiloride followed by combined molecular, cellular, and electrophysiological analyses. We found that (i) *Tmprss2* is highly co-expressed with ENaC in the distal mouse nephron, (ii) αENaC mRNA transcript expression follows *Tmprss2* gene expression, (iii),

loss of *Tmprss2* abolishes ENaC activity, and (iv) Tmprss2 deficiency results in a loss of EpCAM and claudin-3 and claudin-7, which results in impaired transepithelial transport and paracellular barrier. Taken together, our data identify Tmprss2 as a critical protease in the maintenance of the tight junction barrier and regulation of ENaC.

# Results

### *Tmprss2* and *Scnn1a* are co-expressed within the mouse kidney

To investigate the potential relationship between Tmprss2 and ENaC, the expression levels of *Tmprss2* and *Scnn1a* were analysed by quantitative RT–PCR in various organs from male and female C57BL/6J WT mice. This unveiled a similar gene expression pattern between *Scnn1a* (Fig 1A) and *Tmprss2* (Fig 1B) across various organs. The lowest mRNA transcript expression of both genes was found in the brain and heart, whereas the lungs, kidneys, and salivary glands showed the highest expression (Fig 1A and B). We next investigated the spatial distribution and intensity of *Tmprss2* and *Scnn1a* (Fig 2A–D) compared with *Furin* and *Scnn1a* expression (Fig 2E–H), within the kidney using RNAscope in situ hybridization. *Tmprss2* expression is observed throughout the kidney with particular enrichment in the cortex and the papillary transitional epithelium with no signal in the negative control (Figs 2A and S1). Overall, 57% of the cortex cells were identified as single *Tmprss2*–positive, 1% as single *Scnn1a*–positive (SP), 26% as double-positive (DP), and 16% as double-negative (DN) cells (Fig 2B, left panel). A higher proportion of *Tmprss2* SP cells was observed in the kidneys of male mice compared with females in the cortex (Fig 2D, left panel). In the medulla, most of the cells expressed both *Tmprss2* and *Scnn1a* (Fig 2C). *Tmprss2* is strongly expressed in the same tubular regions as *Scnn1a* in the cortex (Fig 2A), and the quantification of *Tmprss2* intensity reveals that there is a threefold increase in DP cells compared with *Tmprss2* SP cells (Fig 2D). *Furin* is expressed throughout the kidney but most highly in the cortex (Fig 2E). Like *Tmprss2*, most of the cells in the cortex are

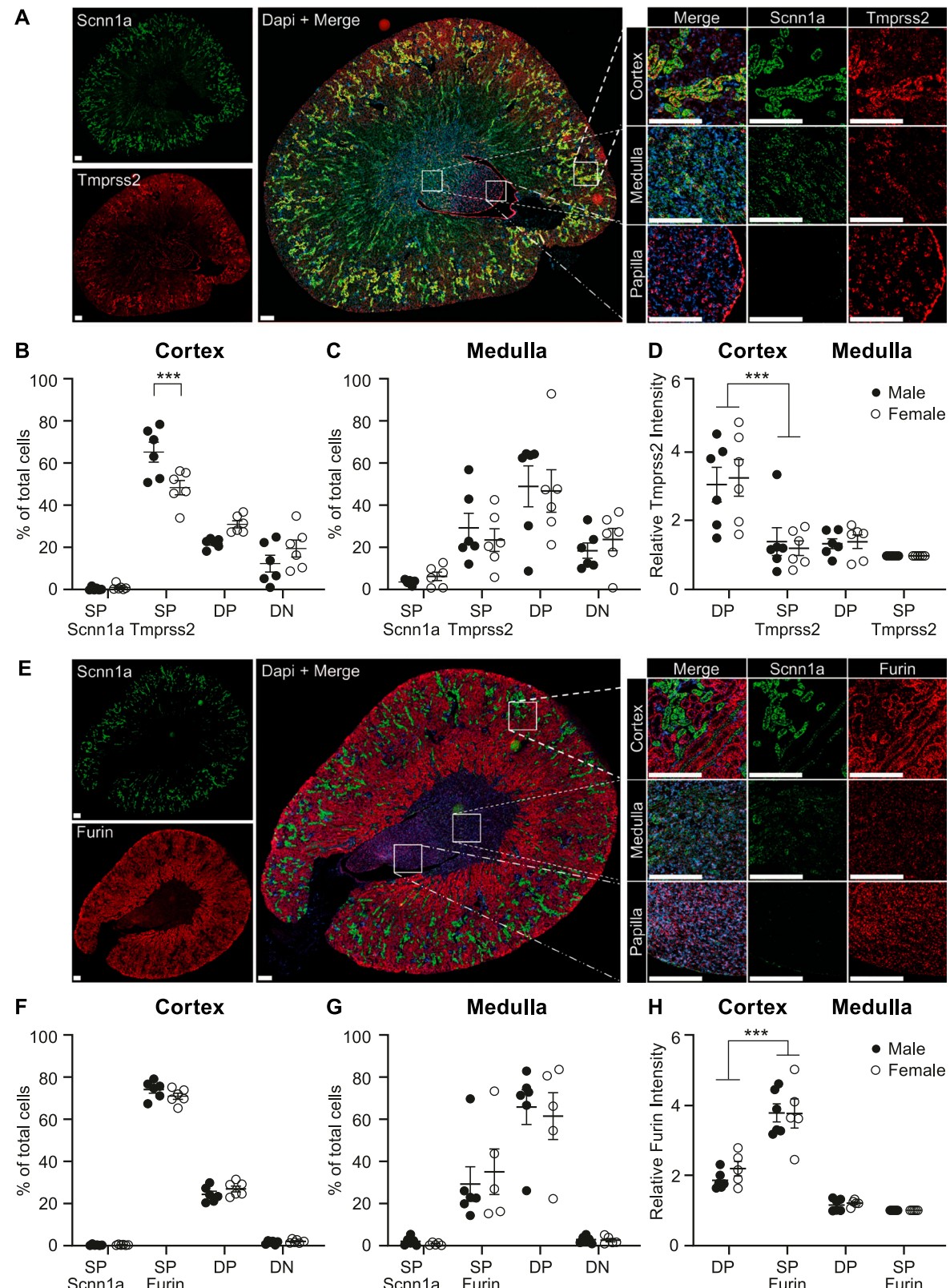

**Figure 2. _Tmprss2_ is highly expressed in _Scnn1a_-positive regions in the cortex, whereas _Furin_ is highly expressed in _Scnn1a_-negative cortex regions.**
**(A, E)** mRNA transcript expression visualized by RNAscope hybridization with probes for (A) _Scnn1a_ (green, upper left), _Tmprss2_ (red channel, bottom left), and _Scnn1a_-_Tmprss2_ merged picture with DAPI counterstain (middle) in the cortex, medulla, and papilla (right panels). **(B, C, D)** Quantification of total cells double-positive (DP) for both _Scnn1a_ and _Tmprss2_, single-positive (SP), or double-negative (DN) in (B) the cortex and (C) medulla. **(D, H)** Relative (D) _Tmprss2_ and (H) _Furin_ intensity in DP and SP of

*Furin* SP (Fig 2F), and in the medulla DN (Fig 2G). In contrast to *Tmprss2*, the highest expression of *Furin* is observed in *Scnn1a*-negative tubular structures (Fig 2E), and the intensity of *Furin* expression is found to be twice as high in *Furin* SP cells compared with *Scnn1a* and *Furin* DP cells (Fig 2H).

In summary, although both *Tmprss2* and *Furin* are expressed throughout the kidney, *Tmprss2* is highly expressed with *Scnn1a* in principal cells of the cortex, whereas *Furin* is highly expressed in *Scnn1a*-negative cells.

### Aldosterone-stimulated *Scnn1a* transcription is accompanied by up-regulated *Tmprss2* gene expression

To study whether Tmprss2 is involved in the regulation of ENaC, we applied aldosterone and amiloride to mCCD$_{cl1}$ wild-type cells for 24 h and analysed the transcription and translation of the Scnn1a and Scnn1g subunits of ENaC and the membrane-bound proteases Furin and *Tmprss2*. To stimulate $\alpha$ENaC maximally, a concentration of 30 and 300 nM aldosterone was induced to observe an increase in Scnn1a mRNA transcript and protein levels, whereas no change was observed in Scnn1g expression (Fig 3A, B, and E). Interestingly, aldosterone stimulation induced *Tmprss2* mRNA transcript up-regulation, whereas no change was observed in *Furin* gene expression despite an increase in protein abundance (Fig 3A, B, and E). The application of amiloride up-regulated *Scnn1a*, but not *Scnn1g* transcription (Fig 3C), accompanied by down-regulated abundance of Scnn1a protein, but not altered Scnn1g protein abundance (Fig 3D and E). Like *Scnn1a*, an up-regulation of *Tmprss2* was induced by amiloride treatment, but no change in Furin gene or protein expression was observed (Fig 3C–E). The application of aldosterone resulted in an increase in ENaC current measured at 5-, 24-, and 48-h post-treatment with a maximal response recorded at 5-h post-treatment, whereas amiloride treatment resulted in a sustained loss of ENaC current (Fig 3F).

To investigate whether there is a specific effect of Scnn1a abundance on the mRNA transcript expression of *Tmprss2*, we generated a lentivirus-transduced *Scnn1a*-inducible cell line. *Tmprss2* gene expression was up-regulated in a doxycycline dose-dependently manner (Fig 4A). The gene and protein expression of Scnn1g and Furin remained unchanged (Fig 4A–C). Wild-type mCCD$_{cl1}$ cells treated or not with doxycycline showed no difference in *Tmprss2*, *Scnn1a*, *Scnn1g*, and *Furin* transcript expression (Fig S2). ENaC current was not changed (Fig 4D). Silenced Scnn1a expression did not alter the mRNA transcript of *Tmprss2* (data not shown).

Overall, the up-regulation of *Scnn1a* gene expression is accompanied by increased *Tmprss2* mRNA levels, whereas Scnn1a silencing does not influence *Tmprss2* gene expression, indicating that Scnn1a followed the *Tmprss2* expression.

### Tmprss2 deficiency results in a decrease in Scnn1a protein expression and function

We next treated wild-type mCCD$_{cl1}$ cells with the unspecific serine protease inhibitors aprotinin and camostat mesylate to inhibit Tmprss2. ENaC current was measured each hour post-treatment for 5 h, which revealed a 37% (10 $\mu$M) and 41% (50 $\mu$M) decrease in current after 1 h of aprotinin, and a 26% (10 $\mu$M) and 31% (100 $\mu$M) decrease after 1 h of camostat mesylate treatment (Fig S3A). Although the mRNA transcript and protein expression of Scnn1a was unchanged, both aprotinin and camostat mesylate treatment resulted in a dose-dependent increase in *Tmprss2* (Fig S3B–D). We suggest that the up-regulation of *Tmprss2* mRNA transcripts may represent a compensatory mechanism for Tmprss2 inhibition, similar to the up-regulation of *Scnn1a* when ENaC is inhibited by amiloride.

To further understand the functional relationship between Tmprss2 and Scnn1a, we used CRISPR/Cas9 gene editing to develop Tmprss2 knockout cell clones. After transfection with two Cas9 guides targeting *Tmprss2*, single cells were isolated by FACS, in addition to control cell clones, and expanded into colonies. These colonies were validated by sequencing and RT–PCR using combinations of primers designed over the region where the Cas9-induced double-stranded breaks occurred revealing near-zero expression (Fig S4A–C). Clones were tested by RT–PCR to ensure the absence of *Slc26a4* (pendrin) expression indicative of an intercalated cell type (Fig S4D). Two knockout (KO1 and KO2) and two control (C1 and C2) clones were selected and used further for experimental testing.

The knockout of *Tmprss2* resulted in a reduced Scnn1a mRNA transcript and protein abundance level. *Scnn1g* mRNA transcript levels but not protein abundance were increased (Fig 5A–C). Furin expression remained unchanged (Fig 5A and B). Interestingly, the voltage, resistance, and ENaC current were reduced in *Tmprss2* knockout clones (Fig 5D–F).

In summary, after *Tmprss2* knockout, aldosterone treatment and Scnn1a ($\alpha$ENaC) overexpression, and amiloride block of ENaC, $\alpha$ENaC mRNA and protein expression followed the *Tmprss2* mRNA expression. As a result of Tmprss2 deficiency, the resistance is near-abolished. Tmprss2-deficient cells do not efficiently form tight epithelia, also explaining the lack of ENaC current.

### RNA-seq analysis of Tmprss2-deficient mCCD cell clones revealed down-regulation of CAP3/St14, the tight junction proteins claudin-3 and claudin-7, and the adhesion molecule EpCAM

To understand the reduced ENaC current in Tmprss2-deficient cells, RNA sequencing was performed to elucidate other implicated genes and to explore their cellular consequences. RNA samples

---

the cortex and medulla from male and female mice. **(E)** RNAscope analysis using probes for *Scnn1a* (green, upper left), *Furin* (red, bottom left), and *Scnn1a-Furin* merged picture with DAPI counterstain (middle) with insets for cortex, medulla, and papilla (right panels). **(F, G, H)** Quantification of total cells double-positive (DP) for both *Scnn1a* and *Furin*, single-positive (SP), or double-negative (DN) in (F) the cortex, (G) medulla, and (H) cortex and medulla from male and female mice. 12 kidneys (six cut transversally and six cut longitudinally) from three male and three female C57BL/6J mice were analysed. Scale bar: 200 $\mu$m. Experiments are represented as the mean ± SEM. Data were analysed by an unpaired two-tailed Welch *t* test, and *P*-values < 0.05 were considered statistically significant; ***$P$ < 0.001. Source data are available for this figure.

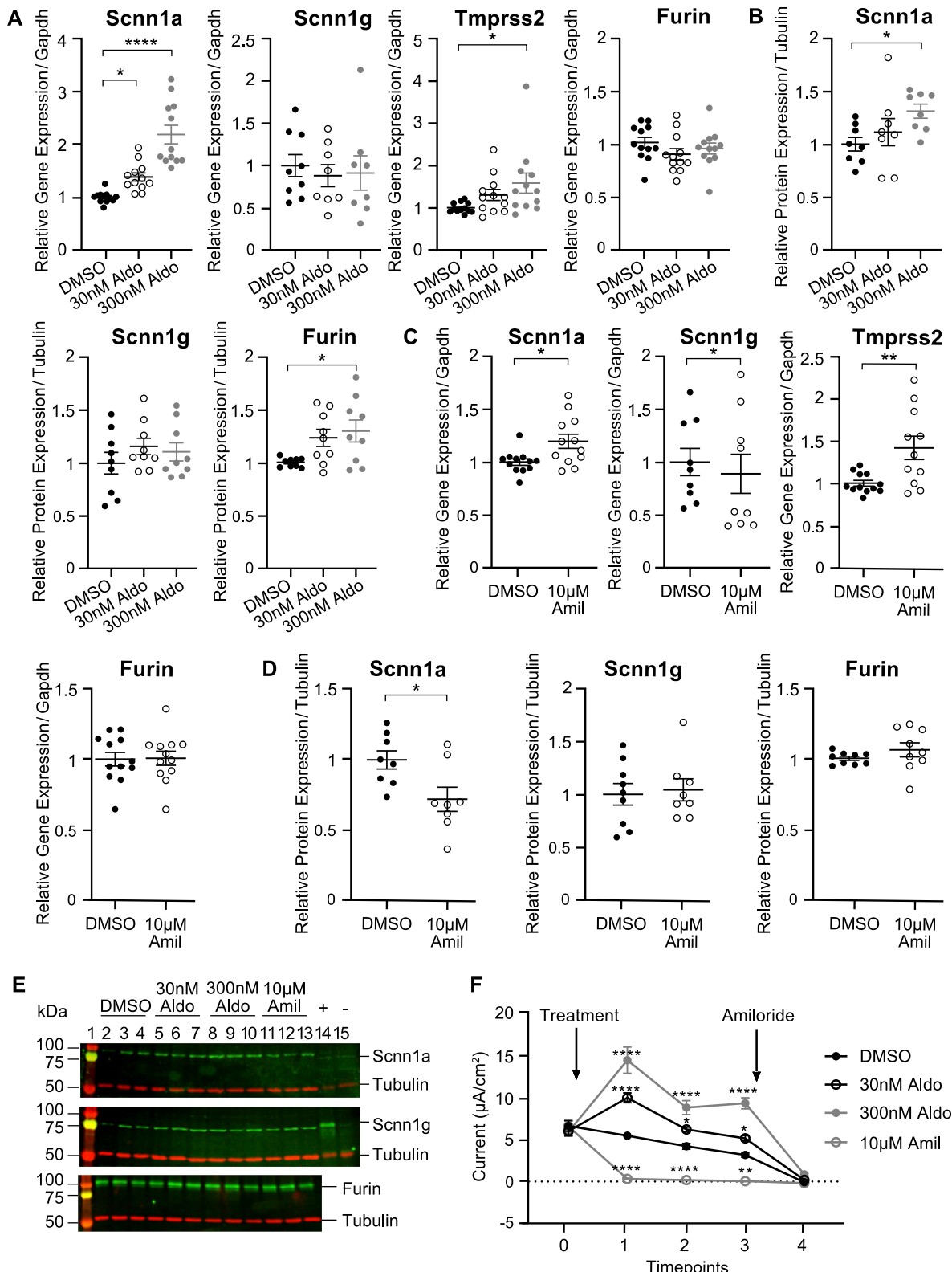

**Figure 3. Aldosterone and amiloride stimulated the mRNA transcript expression of *Scnn1a* and *Tmprss2*, despite opposing effects of aldosterone and amiloride on Scnn1a protein expression and ENaC current.**
**(A, B, C, D)** Gene (A, C) and (B, D) protein expression of Scnn1a, Scnn1g, *Tmprss2*, and Furin in response to 30 nM and 300 nM aldosterone (A, B), and 10 μM amiloride (C, D) treatment, normalized to (A, C) *Gapdh* and (B, D) tubulin expression; n = 12, triplicate samples from four independent experiments after 24-h treatment. **(E)** Representative Western blot analysis of the protein expression of Scnn1a and Scnn1g, and Furin after treatment of aldosterone and amiloride, normalized to tubulin. Values are

from two control and two *Tmprss2* knockout clones derived from three independent experimental replicates from three different passages were analysed. The principal component analysis (Fig 6A) and genetic heatmap (Fig 6B) show that there is a high degree of correlation between the two control clones and the two *Tmprss2* knockout clones. A similar number of genes were found to be differentially up- and down-regulated in Tmprss2 knockout compared with control cells (Fig 6C). To identify strongly differentially expressed genes, comparisons between control and *Tmprss2* knockout clones were performed and genes with a fold change of 2, a false discovery rate (FDR) of 0.05, and an adjusted *P*-value of < 0.05 present in the gene intersection were considered as differentially expressed genes. The gene intersection common to four comparisons (Fig 6D) revealed 669 differentially expressed genes, 323 up-regulated and 345 down-regulated (one gene regulated in opposite directions in different comparisons), in knockout compared with control clones. Reactome pathway analysis of the 668 genes revealed that differentially regulated pathways belong to membrane trafficking and transport (Fig 7A). Next, we looked at specific genes, which showed that many tight junction–related genes were differentially regulated including *Epcam, Cldn2, Cldn3, Cldn7,* and *Cldn23* (Fig 7B). Serine proteases, for example, *St14* (matriptase), and their inhibitors *Spint1* and *Spint2* were down-regulated in *Tmprss2* knockout cells (Fig 7B). Transcription and translation of CAP1/Prss8 (prostasin) was not affected (Fig S5A and B). CAP2/*Tmprss4* and *Tmprss1* (hepsin) were not detected in the RNA-seq data (Fig 7). Tight junction–associated genes such as *Cldn4, Cldn8,* and *Tjp1* (ZO-1), *Ocln* (occludin), and *Cdh1* (E-cadherin) were not differentially regulated in *Tmprss2* knockout cells neither on the mRNA transcript (Fig S5C, E, and G) nor on the protein (Fig S5D, F, and H) level.

To validate the RNA-sequencing findings, RT–PCR, Western blot analyses, and immunocytochemistry of EpCAM, claudin-2, claudin-3, and claudin-7 were performed. Indeed, the gene expression of *Epcam* and of *Cldn3* and *Cldn7* was drastically down-regulated; however, *Cldn2* mRNA transcript levels were unchanged and low (Fig 8A). The protein abundance of EpCAM, claudin-3, and claudin-7 was also found to be down-regulated in Tmprss2 knockout cells (Fig 8B and C) and near-absent in immunocytochemistry (Fig 8G). In addition to the low *Cldn2* gene expression, there was no visible claudin-2 protein expression in the control or knockout cells (Fig 8B, left panel). We tested the mRNA (Fig 8D) and protein abundance (Fig 8E and F) of the membrane-bound serine protease CAP3/St14 (matriptase), which was reduced in *Tmprss2* knockout cells (Fig 8D and E). Modulation of ENaC itself either by aldosterone and amiloride treatment (Fig S6A–C) or by direct Scnn1a up-regulation (Fig S6D) or silencing (Fig S6E and F) did not affect the expression of the adhesion molecule EpCAM and the tight junction proteins claudin-3 and claudin-7.

To summarize, Tmprss2 deficiency results in a drastic reduction in the gene and protein expression of the alpha subunit of ENaC, which subsequently leads to loss of transepithelial sodium transport as evidenced by near-abolished ENaC current because of near-zero resistance and voltage. This indicates that Scnn1a expression is highly dependent on Tmprss2. The observation that Scnn1a up-regulation induces *Tmprss2* up-regulation further suggests that Scnn1a requires Tmprss2, whereas Scnn1a silencing does not alter *Tmprss2* mRNA transcript levels. In addition to its regulatory role of *Scnn1a*, Tmprss2 deficiency affected the epithelial barrier structure and function by drastically down-regulating the adhesion molecule EpCAM and claudin-3 and claudin-7, which regulate the paracellular permeability. These data indicate that Tmprss2 has a dual role in maintaining the function of ENaC by regulating *Scnn1a* expression and maintaining the epithelial tight junction barrier likely through its role in regulating EpCAM and subsequently claudin-3 and claudin-7 expression and/or localization.

# Discussion

### Regulation of the ENaC-mediated transepithelial sodium transport by Tmprss2 via CAP3/St14

ENaC is a transmembrane channel crucial in maintaining sodium balance and fluid homeostasis in various epithelial tissues, such as the kidney, lung, and colon (Ehret & Hummler, 2022). In this study, using the well-established mCCD$_{cl1}$ cell line, we investigated the relationship between Tmprss2 and ENaC by modulating their expression. The expression profiles of the mRNA transcript expression of *Tmprss2* and *Scnn1a* were quantitatively and qualitatively quite similar in various tissues (Fig 1), and the co-expression and co-localization was confirmed in mouse kidney sections using RNA-scope technology (Fig 2). *Scnn1a* mRNA transcripts are primarily expressed in the second part of the distal convoluted duct (DCT2), connecting tubule (CNT), and the cortical collecting duct (CCD) (Chen et al, 2021). *Tmprss2* expression followed the Scnn1a gene and protein expression when stimulated by aldosterone- or doxycycline-induced Scnn1a gene and protein expression (Figs 3 and 4). Two former studies reported a 2.6-fold (Faller et al, 2014) and threefold (Sure et al, 2022) increase in ENaC current when *Tmprss2* was co-injected with ENaC subunits into the *Xenopus* oocytes. However, we found that *Tmprss2* expression is linked to *Scnn1a* rather than to *Scnn1g*. Furthermore, our data rather confirm initial studies, which reported markedly decreased ENaC current and protein levels when ENaC subunits were co-injected with *Tmprss2* into *Xenopus* oocytes (Donaldson et al, 2002). It can, however, not be excluded that the

expressed relative to normalized DMSO (vehicle) values. Kidney lysates from control and renal tubule–specific knockout mice of Scnn1a (Perrier et al, 2016) and Scnn1g (Boscardin et al, 2018) were used as positive (+) and negative (−) controls for the antibodies. **(F)** Time course of ENaC current of mCCD$_{cl1}$ cells treated with DMSO control (closed black circles), 30 nM aldosterone (open black circles), 300 nM aldosterone (closed grey circles), and 10 $\mu$M amiloride (open grey circles). Cells treated after first measurement ($t_{0h}$), measured at timepoints 1 (5 h), 2 (24 h), and 3 (48 h) post-treatment; at $t_{48h}$, amiloride (10 $\mu$M, arrow) was added to the apical side of all filters for 10 min (timepoint 4); n = 8 from three independent experiments that are represented as the mean ± SEM with significance determined at each timepoint compared with DMSO (vehicle) control. Note that x-axis is not drawn to scale. **(A, B, C, D, F)** Data were analysed by either an unpaired *t* test (C, D), or one-way (A, B) or two-way (F) ANOVA with Dunnett's multiple comparison test comparing values with DMSO. *P*-values < 0.05 were considered as statistically significant; *$P$ < 0.05, **$P$ < 0.01, and ****$P$ < 0.0001. Source data are available for this figure.

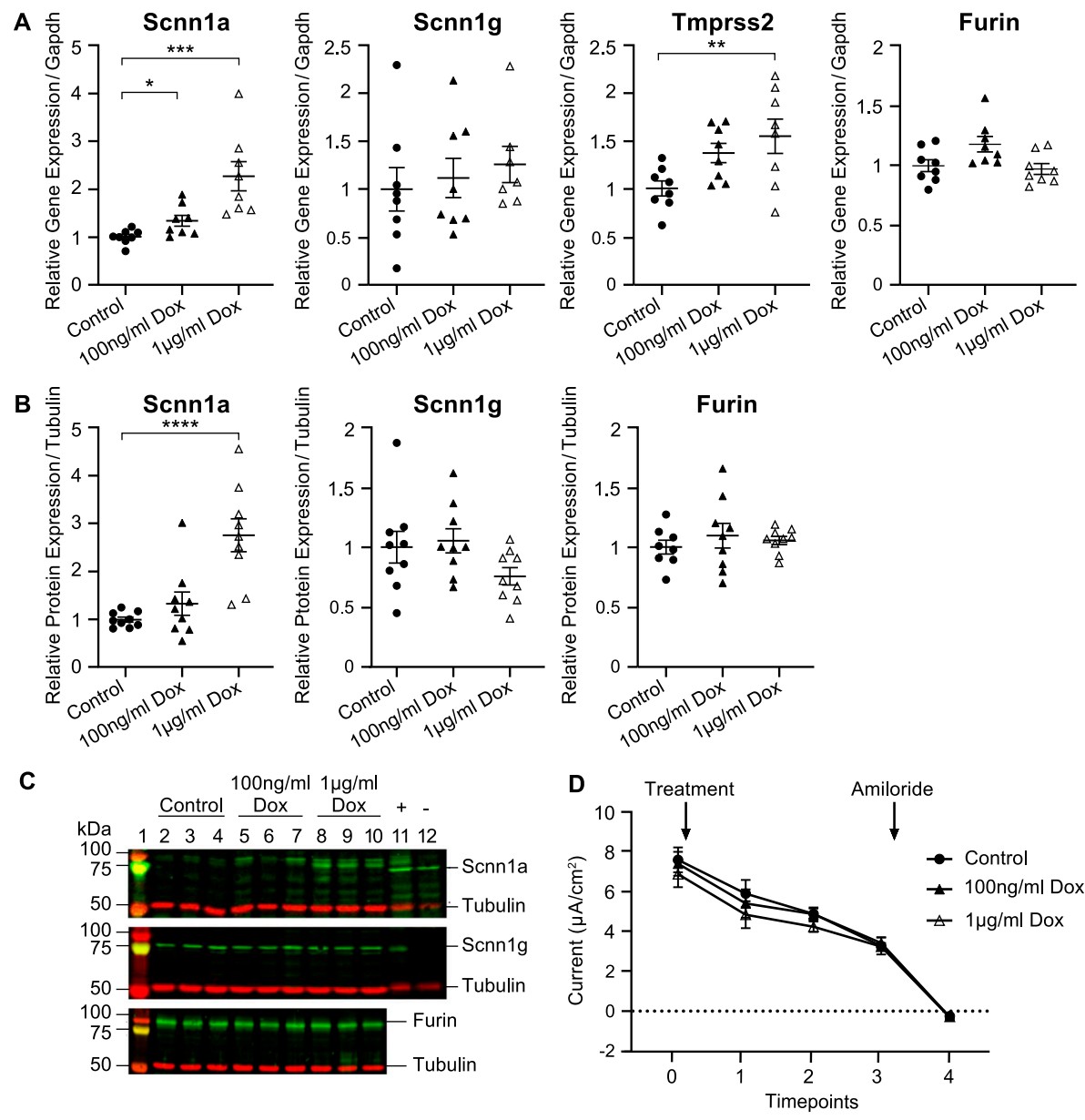

**Figure 4. Doxycycline-induced Scnn1a overexpression is accompanied by up-regulated *Tmprss2* gene expression.**
Lentivirus-transduced mCCD$_{cl1}$ cell clones encoding a doxycycline-inducible αENaC-Tet-One plasmid to induce Scnn1a overexpression with either 100 ng/ml and 1 μg/ml of doxycycline, or no doxycycline as a control. **(A)** Gene expression of *Scnn1a*, *Scnn1g*, *Tmprss2*, and *Furin* normalized to *Gapdh*. **(B)** Quantification of the protein expression of Scnn1a, Scnn1g, and Furin normalized to tubulin after 24-h treatment. **(C)** Representative Western blot analysis (n = 7–9 from three independent experiments that are represented as the mean ± SEM. Kidney lysates from control and renal tubule–specific knockout mice of Scnn1a (Perrier et al, 2016) and Scnn1g (Boscardin et al, 2018) were used as positive (+) and negative (−) controls for the antibodies. **(D)** Time course of ENaC current of Scnn1a-induced mCCD$_{cl1}$ cells where Scnn1a is ± induced after the first measurement ($t_{0h}$), then measured at timepoints 1 (5 h), 2 (24 h), and 3 (48 h) post-treatment; at timepoint 3 ($t_{48h}$), amiloride (10 μM, arrow) was added to the apical side of all filters for 10 min and measured at timepoint 4. Note that x-axis is not drawn to scale; n = 8 from 3 independent experiments that are represented as the mean ± SEM. **(A, B, D)** Data were analysed by one-way (A, B) or two-way (D) ANOVA with Dunnett's multiple comparison test comparing values with controls. *P*-values < 0.05 were considered as statistically significant; *P < 0.05, **P < 0.01, ***P < 0.001, and ****P < 0.0001.
Source data are available for this figure.

different outcomes of the *Xenopus*-derived data might be due to species specificity of the constructs, dependency of dose, and/or time of expression (Grant & Lester, 2021). On the contrary, *Scnn1a* shRNA silencing in mCCD cells resulted in decreased Scnn1a mRNA transcript levels and protein abundance, which did not affect *Tmprss2* expression (data not shown; Sassi et al, 2020).

Using aprotinin and camostat mesylate, two non-specific serine protease inhibitors, we measured a decrease in ENaC current, despite unchanged Scnn1a gene and protein expression, but increased *Tmprss2* mRNA transcript levels (Fig S3). This is in agreement with previous measurements obtained from primary cultures of human nasal airway epithelial cells after aprotinin treatment,

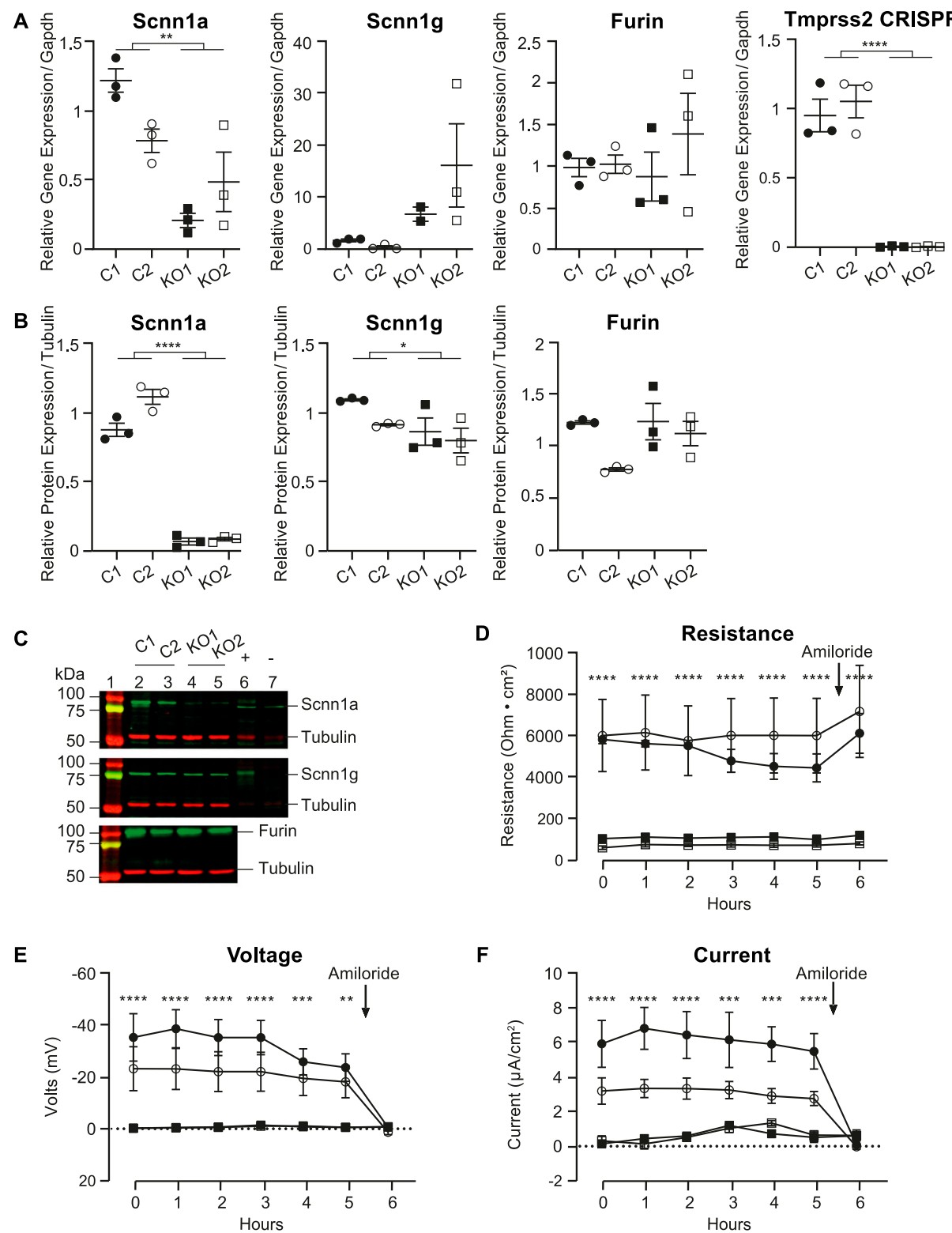

**Figure 5.** *Tmprss2* **deficiency results in reduced Scnn1a mRNA transcript expression, protein abundance, and amiloride-sensitive ENaC current.**
**(A, B)** Quantitative mRNA transcript expression of *Scnn1a*, *Scnn1g*, *Furin*, and *Tmprss2* (using primers overlapping CRISPR/Cas9 cleavage site) normalized to *Gapdh*, and (B) the protein expression of Scnn1a, Scnn1g, and Furin normalized to tubulin from control and Tmprss2 knockout clones; kidney lysates from control and renal tubule–specific knockout mice of Scnn1a (Perrier et al, 2016) and Scnn1g (Boscardin et al, 2018) were used as positive (+) and negative (−) controls. Data were obtained from two control and two Tmprss2 knockout cell clones. Each dataset was performed in triplicate, and values are expressed relative to control colonies. **(C)** Representative Western blot analyses are shown. **(D, E, F)** Time course (6 h) experiment of ENaC current of control and Tmprss2 knockout cells measuring resistance (Ohms × cm²), voltage

which resulted in markedly declined ENaC current and Scnn1a protein expression levels (Donaldson et al, 2002). In the same line, *Xenopus* oocytes and the human airway epithelial cell line H441 still showed a substantial preserved ENaC current after aprotinin treatment (Sure et al, 2022), which was explained by an incomplete protease inhibition and thus partial ENaC activation. In vitro, several serine proteases have been identified as ENaC/channel-activating proteases (CAPs) (Anand et al, 2022), which likely differ in their aprotinin sensitivity, for example, CAP3/St14 (matriptase), which was reported resistant (Vuagniaux et al, 2002). It is interesting to note that aprotinin treatment blocked ENaC-mediated sodium retention in a mouse model of experimental nephrotic syndrome. It mimicked the block of ENaC by its specific inhibitor, amiloride (Bohnert et al, 2018). In the lack of specific serine protease inhibitors and the presence of multiple proteases, it is difficult to identify the relevant one(s) regulating ENaC. The protease composition in human airway cells likely also differs from the renal mCCD$_{cl1}$ cell line explaining certain variability by blocking ENaC function using protease inhibitors.

### Tmprss2 deficiency affected the epithelial integrity and is accompanied by loss of various tight junction proteins

Tmprss2 deficiency resulted in reduced ENaC current, which can be explained by the near-complete absence of Scnn1a mRNA transcripts and protein (Fig 5). Scnn1g protein abundance was reduced, whereas *Scnn1g* mRNA transcript levels were increased, which may be explained as the cells compensate for the reduction in Scnn1a and Scnn1g protein expression. The expression of the serine protease Furin was not altered (Fig 5A–C). *Scnn1a* gene expression followed the *Tmprss2 mRNA* transcript expression. We cannot, however, exclude that Tmprss2 partly regulates ENaC via the channel-activating protease CAP3/St14 that was drastically down-regulated in *Tmprss2* knockout cells (Fig 8D and E).

The RNA-sequencing analyses further revealed that several tight junction proteins were also affected in their expression (Figs 6 and 7). Amongst those, claudin-3 and claudin-7 and EpCAM were validated by RT–PCR and Western blot analyses (Fig 8A–C). Two proteases, namely, the serine protease *Prss22* (Wong et al, 2001) belonging to the trypsin family of serine proteases, and *St14* (CAP3/matriptase) (Lin et al, 1999), were reduced (Figs 6 and 7). It has been reported that the apical treatment of serine proteases to the canine epithelial cell line SCBN caused a rapid and sustained increase in transepithelial electrical resistance (TER) (Ronaghan et al, 2016). The reduction in *St14* gene expression in our Tmprss2-deficient cell clones may thus contribute to the near-abolished transepithelial resistance (Fig 5). Indeed, constitutive knockout mice deficient for St14, similar to epidermis-specific *Prss8* (CAP1/prostasin) knockout mice, presented with impaired epithelial barrier functions (List et al, 2002; Leyvraz et al, 2005). This was accompanied by a complete loss

of the tight junction protein occludin in *Prss8* knockout mice (Leyvraz et al, 2005) and its mislocation in *St14* knockout mice (List et al, 2002).

Recently, a novel mechanism of tight junction maintenance through localized proteolysis of EpCAM at tight junction leaks was proposed to explain how the epithelial barrier is regulated in claudin-2–deficient MDCK II cells (Higashi et al, 2023). The data from the Tmprss2-deficient mCCD$_{cl1}$ cell line are compatible with this mechanism by which Tmprss2 accesses the EpCAM/claudin-7 complex at breached tight junctions and cleaves EpCAM, thereby liberating claudin-7. Tmprss2 deficiency in mCCD$_{cl1}$ cells reduces the expression of EpCAM as evidenced by molecular and immunofluorescence analyses (Fig 8). As a consequence, claudin-3 and claudin-7 are reduced in these cells (Fig 8D). Slug and Twist1 are transcription factors implicated in the negative regulation of the EpCAM promoter (Liu et al, 2021), but no or only weak expression was detected in *Tmprss2* control and knockout cells (data not shown). The down-regulation of the two serine proteases *St14* and *Prss22* suggests that Tmprss2 acts upstream leading to a complete loss of resistance and voltage, and therefore current (Fig 5). In addition, several proteases may participate in the same cell. This could also explain why Higashi and co-workers only found a transient loss of TER in MDCK cells deficient for St14 or Prss8 (Higashi et al, 2023). Furthermore, Wu and co-workers (2020) showed that matriptase (St14) cleaves EpCAM and silencing the serine protease inhibitors HAI-1 (*Spint1*) and HAI-2 (*Spint2*) leads to a drastic decrease in matriptase, claudin-7, and EpCAM full-length protein expression, despite an increase in EpCAM cleavage (Wu et al, 2020). In line with these results, a reduction in full-length EpCAM protein expression is observed in our data upon *Tmprss2* knockout, and an up-regulation in *Spint1* expression, further suggesting that Tmprss2 is involved (Wettstein et al, 2022) in the same regulatory tight junction pathway as the serine protease matriptase.

In vivo, the same regulatory mechanisms are not observed in all organs; for example, *Prss8* knockout mice showed an impaired barrier function resulting in severe dehydration affecting tight junctions in the epidermis (Leyvraz et al, 2005), whereas the colon-specific *Prss8* knockout revealed a defect in ENaC-mediated transepithelial sodium transport without impaired epithelial integrity (Malsure et al, 2014). St14-deficient mice, however, showed an impaired integrity of tight junctions in the skin and colon (List et al, 2002; Buzza et al, 2010), indicating protease- and organ-specific effects.

In the mouse cortical collecting duct kidney cells, our data clearly indicate that Tmprss2 deficiency affects transepithelial sodium transport in addition to epithelial barrier integrity by depleting the EpCAM/claudin-7 complex leading to a complete loss of resistance (Fig 5). The modification of ENaC expression and function does not alter the gene and protein expression of the EpCAM/claudin-7 complex (Fig S6).

---

(Volts, mV), and current ($\mu A/cm^2$). At $t_{5h}$, amiloride (10 $\mu M$, arrow) was added to the apical side of all filters for 10 min; n = 5 from three independent experiments that are represented as the mean ± SEM. **(A, B, D, E, F)** Data were analysed by an unpaired *t* test (A, B) or two-way ANOVA (D, E, F) with Sidak's multiple comparison test. *P*-values < 0.05 were considered as statistically significant; *$P < 0.05$, **$P < 0.01$, ***$P < 0.001$, and ****$P < 0.0001$. Source data are available for this figure.

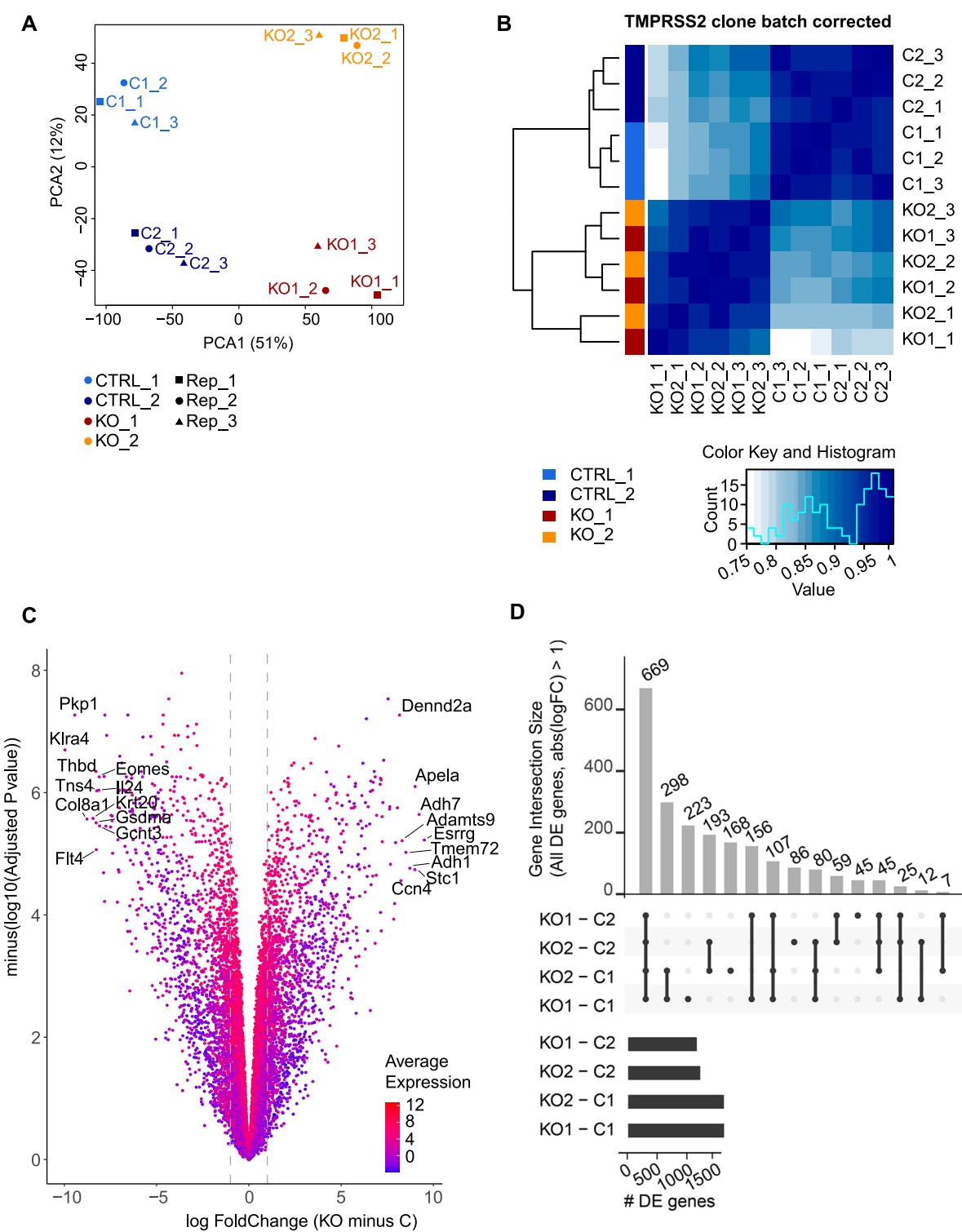

**Figure 6. RNA-seq data of *Tmprss2* knockout revealed differentially regulated tight junction–related genes.**
**(A)** Principal component analysis of RNA-seq gene expression data from three replicates of two control (light and dark blue) and two *Tmprss2* knockout (orange and red) clones from n = 3 passages (−1, −2, and −3). **(B)** Heatmap of sample-to-sample genetic correlation. Correlations were calculated using the Pearson correlation coefficient. **(C)** Volcano plot of differentially expressed genes in *Tmprss2* knockout compared with control cells, the x-axis represents the log₂ fold change, and the y-axis represents minus log 10 of the adjusted *P*-value with dot colour indicating the average gene expression of the two conditions. **(D)** Upset plot of differentially expressed genes (FC > 2). Vertical bars show the number of commonly detected genes between the analysed groups.

**A**

| ID | Reactome Pathway | Count | p-value |
|---|---|---|---|
| R-MMU-6809371 | formation of the cornified envelope | 12 | 9.16E-09 |
| R-MMU-199991 | membrane trafficking | 29 | 3.41E-08 |
| R-MMU-4085001 | Sialic Acid metabolism | 8 | 3.83E-08 |
| R-MMU-392499 | metabolism of proteins | 56 | 9.02E-08 |
| R-MMU-382551 | transport of small molecules | 31 | 1.09E-07 |
| R-MMU-597592 | post translational protein modification | 47 | 1.81E-07 |
| R-MMU-1430728 | metabolism | 56 | 6.53E-07 |
| R-MMU-983712 | Ion channel transport | 14 | 7.27E-07 |
| R-MMU-1266738 | developmental biology | 24 | 9.64E-07 |
| R-MMU-5653656 | vesicle mediated transport | 30 | 1.20E-06 |
| R-MMU-380108 | chemokine receptors bind chemokines | 8 | 2.36E-06 |
| R-MMU-109582 | Hemostasis | 27 | 2.88E-06 |
| R-MMU-8856825 | cargo recognition for Clathrin mediated endocytosis | 10 | 5.05E-06 |
| R-MMU-195721 | signaling by WNT | 15 | 5.40E-06 |
| R-MMU-446203 | Asparigine N linked Glcosylation | 16 | 5.61E-06 |
| R-MMU-446219 | synthesis of substrates in N-Glycan biosythesis | 8 | 5.87E-06 |
| R-MMU-9006931 | signaling by nuclear receptors | 14 | 6.73E-06 |
| R-MMU-211859 | biological oxidations | 14 | 7.09E-06 |
| R-MMU-212718 | EGFR interacts with phospholipase C-Gamma | 4 | 7.54E-06 |
| R-MMU-9009391 | extra nuclear Estrogen signaling | 8 | 1.17E-05 |
| R-MMU-6805567 | Keratinization | 12 | 1.90E-05 |
| R-MMU-5683057 | MAPK family signaling cascades | 16 | 2.10E-05 |
| R-MMU-9006934 | signaling by receptor Tyrosine Kinases | 19 | 2.19E-05 |
| R-MMU-2672351 | stimuli sensing channels | 9 | 2.48E-05 |
| R-MMU-1474244 | extracellular matrix organization | 14 | 3.72E-05 |
| R-MMU-180336 | SHC1 events in EGFR signaling | 4 | 4.01E-05 |

**B**

| Gene | Description | Log fold change (KO minus C) | Log10 adjusted p Value |
|---|---|---|---|
| **Tight Junctions and ENaC** | | | |
| Cldn7 | claudin 7 [Source:MGI Symbol;Acc:MGI:1859285] | -7.008 | 5.749 |
| Epcam | epithelial cell adhesion molecule [Source:MGI Symbol;Acc:MGI:106653] | -4.438 | 5.268 |
| Cldn3 | claudin 3 [Source:MGI Symbol;Acc:MGI:1329044] | -3.704 | 6.257 |
| Cldn23 | claudin 23 [Source:MGI Symbol;Acc:MGI:1919158] | -3.626 | 4.304 |
| Cldn2 | claudin 2 [Source:MGI Symbol;Acc:MGI:1276110] | 4.546 | 4.116 |
| Scnn1g | sodium channel, nonvoltage-gated 1 gamma [Source:MGI Symbol;Acc:MGI:104695] | 7.903 | 4.185 |
| **Proteases and Inhibitors** | | | |
| Serpinb5 | serine (or cysteine) peptidase inhibitor, clade B, member 5 [Source:MGI Symbol;Acc:MGI:109579] | -7.707 | 6.936 |
| Prss22 | protease, serine 22 [Source:MGI Symbol;Acc:MGI:1918085] | -7.008 | 6.039 |
| Spink2 | serine peptidase inhibitor, Kazal type 2 [Source:MGI Symbol;Acc:MGI:1917232] | -5.736 | 6.908 |
| Spint1 | serine protease inhibitor, Kunitz type 1 [Source:MGI Symbol;Acc:MGI:1338033] | -4.076 | 5.026 |
| St14 | suppression of tumorigenicity 14 (colon carcinoma) [Source:MGI Symbol;Acc:MGI:1338881] | -3.315 | 6.227 |

**Figure 7.  Reactome pathway analysis of Tmprss2-deficient mCCDcl1 cell clones.**
**(A)** Reactome pathway analysis of 669 genes present in the gene intersection with adjusted *P*-values. **(B)** RNA encoding selected tight junction proteins, ENaC, proteases, and protease inhibitors with significantly differential expression are indicated.

**Tmprss2 affects both the transcellular and paracellular transport**

Our data clearly show that Tmprss2 is involved in the regulation of ENaC and epithelial barrier function as its absence leads to a reduction in Scnn1a, EpCAM, and the associated tight junction proteins claudin-3 and claudin-7. Stimulation and inhibition of ENaC by aldosterone or amiloride, and the doxycycline-induced stimulation of Scnn1a do not affect the gene or protein expression of EpCAM,

claudin-3, and claudin-7, indicating that altered ENaC activity does not directly affect the epithelial barrier integrity (Fig S6). Recently, Sassi and co-workers (2020) proposed a close interaction between Scnn1g and claudin-8 that modulates the paracellular sodium permeability in mCCD$_{cl1}$ cells (Sassi et al, 2020). Overexpression and silencing of Scnn1g was associated with altered claudin-8 abundance, and *Scnn1g* knockout mice displayed reduced claudin-8 abundance. Interestingly, Scnn1a did not alter its expression

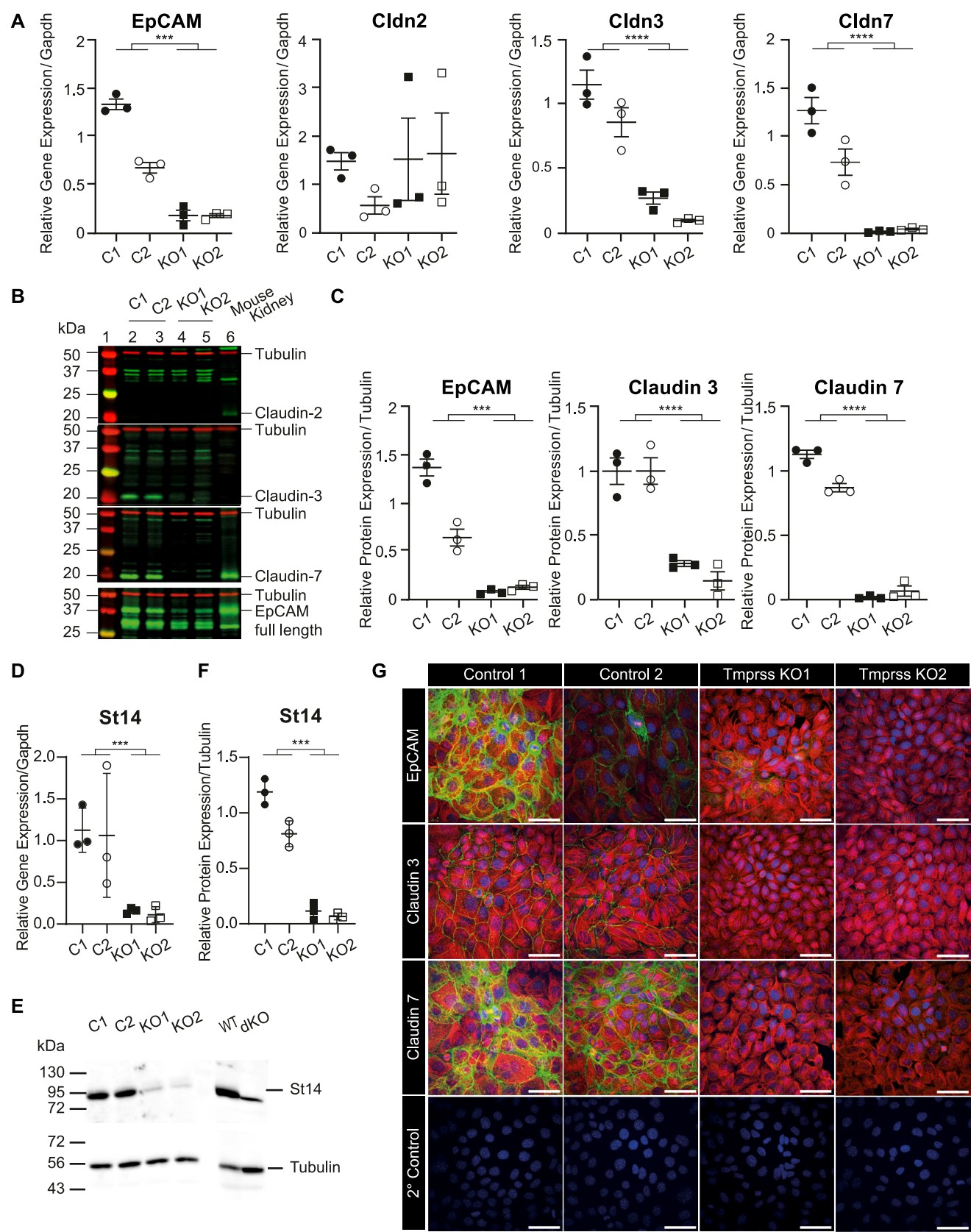

(Sassi et al, 2020). In our study, claudin-8 is not differentially expressed in *Tmprss2* knockout cells (Fig 6), and Scnn1g, albeit increased mRNA transcript expression was rather decreased on a protein level in *Tmprss2* knockout cells (Fig 5), indicating that different ENaC channel subunits exert different effects on tight junction regulation. The *Tmprss2* knockout mice first described by Kim and co-workers (Kim et al, 2006) did not exhibit any obvious phenotype likely because of the fact that the knockout was incomplete. The mice lacked two amino acids of the catalytic domain but still exhibited 80% of the mRNA transcripts. Although not demonstrated in the original paper, these mice might express aberrant proteins and/or a catalytically dead Tmprss2 protein, which could suggest a role of Tmprss2 independent of its catalytic domain as described for the serine protease CAP1/Prss8 (Peters et al, 2014). It will be interesting to analyse the renal tubule–specific phenotype of a complete knockout.

In summary, in mCCD$_{cl1}$ cells, transepithelial sodium transport mediated by ENaC is compromised in cells lacking *Tmprss2*. The epithelial barrier integrity is normally maintained through proteases, ion channels, and tight junctions (Fig 9A). In our working model, we propose that Tmprss2 acts upstream of ENaC, CAP3/St14 (matriptase), and EpCAM and directly affects the EpCAM/claudin-7 co-localization by disrupting the EpCAM/claudin-7 complex, thereby participating in the functional renewal of the tight junction proteins claudin-7 and claudin-3 (Fig 9A). In its absence, this likely results in impaired transepithelial and paracellular transport because of the defective tight junction maintenance and repair mechanism, as evidenced by the near-abolished transepithelial resistance. The tight epithelium becomes leaky (Fig 9B). This is consistent with loss of the expression of selected claudins in mice and humans with EpCAM mutations (Lei et al, 2012; Mueller et al, 2014). Further studies should address whether Tmprss2 indirectly affects the expression of genes by cleaving and activating certain signalling molecules or receptors, which in turn modulate the downstream gene expression of, for example, specific transcription factors. Like many serine proteases, Tmprss2 exerts multiple functions including the facilitation of SARS-CoV-2 entrance together with Furin into host cells (Iwata-Yoshikawa et al, 2022). Impaired ENaC transport in airways of patients with COVID-19 may be thus explained by this dual function of Tmprss2 (Gentzsch & Rossier, 2020). Yet, our study reveals an important and unexpected insight into the hierarchy of specific membrane-bound serine proteases and their activation/inhibition mechanism. Tmprss2 plays a crucial role in the regulation of transcellular and paracellular transport that includes ENaC-mediated sodium transport and EpCAM/claudin-7– mediated tight junction maintenance. Future studies will have to dissect which of the membrane-bound serine proteases are particularly relevant in various cell types.

# Materials and Methods

## mCCD$_{cl1}$ cell culture and transepithelial current measurements

mCCD$_{cl1}$ cells derived from spontaneously transformed mouse CCD primary cultures (Gaeggeler et al, 2005) were maintained at 37°C and 5% CO$_2$ in full growth medium: DMEM/F-12 GlutaMAX (#31331; Gibco), supplemented with 5 g/ml insulin (#I-1882; Sigma-Aldrich), 50 nM dexamethasone (#D-8893; Sigma-Aldrich), 60 nM selenium (#S-9133; Sigma-Aldrich), 5 µg/ml apotransferrin (#T-1428; Sigma-Aldrich), 1 nM triiodothyronine (#T-5516; Sigma-Aldrich), 5 ng/ml mouse EGF (#E-4127; Sigma-Aldrich), 50 U/ml penicillin and 50 µg/ml streptomycin (#15070063; Gibco), and 2% (vol/vol) decomplemented fetal bovine serum (#355500; Corning).

For RNA and protein analyses, 500,000 cells were seeded per well in a six-well plate, and if required treated with relevant drugs after 24 h, then harvested after 48 h. For transepithelial voltage experiments, 1 million confluent mCCD$_{cl1}$ cells were seeded and grown in full growth medium on collagen-treated permeable filters (Transwell, 4.7 cm$^2$, pore: 0.4 µm, polycarbonate membrane; #3412; Corning Costar). On the fifth day, cells were then grown in filter medium (growth medium without dexamethasone, apotransferrin, EGF, and FBS), and on the 10$^{th}$ day, cells were starved with basal DMEM/F-12 GlutaMAX before starting measurements on the 11$^{th}$ day at room temperature. Transepithelial voltage and resistance measurements were performed with a voltohmmeter (Millicell ERS-2; Millipore), and transepithelial current was calculated according to Ohm's law.

## CRISPR/Cas9 gene editing

Two guides were designed in the third exon of the *Tmprss2* gene using Benchling (https://benchling.com/crispr), in addition to one guide targeting luciferase as a control, and ordered as standard oligo pairs from Microsynth (Table 1). In brief, oligo pairs were annealed and phosphorylated; *Tmprss2* sgRNA pair 1 cloned into pU6-(BbsI)_CBh-Cas9-T2A-mCherry (#64324; Addgene); *Tmprss2* sgRNA pair 2 cloned into pSpCas9(BB)-2A-GFP (#48138; Addgene); luciferase control sgRNA pair cloned into both plasmids; correct guide insertion was verified by sequencing using the hU6-F primer 5′-GAGGGCCTATTTCCCATGATTCC-3′. mCCD$_{cl1}$ cells were double-transfected with mCherry and GFP plasmids containing either Tmprss2- or luciferase control–targeting guides using Lipofect-amine 2000 (#11668019; Invitrogen). After 24 h, cells were FACS-single-cell–sorted into a 96-well plate to isolate mCherry and GFP double-positive cell clones.

**Figure 8. Tmprss2 deficiency results in reduced EpCAM and claudin-3 and claudin-7 expression.**
**(A, B, C)** Gene and (A, C) protein expression of EpCAM, claudin-2 (*Cldn2*), claudin-3 (*Cldn3*), and claudin-7 (*Cldn7*) normalized to *Gapdh* and tubulin, respectively. **(B)** Representative Western blots of claudin-2, claudin-3, claudin-7, and EpCAM in two control and two *Tmprss2* knockout cell clones (n = 3 from three independent experiments that are represented as the mean ± SEM). The kidney lysate from a wild-type mouse was used as a protein loading reference. **(B, C)** Quantifications of Western blot data of EpCAM, claudin-3, and claudin-7 in (B). **(D, E, F)** Quantitative mRNA transcript and (E) protein expression and (F) its quantification of St14 in *Tmprss2* control (open and closed circles) and knockout (open and filled squares) cells (n = 3 from three independent experiments that are presented as ± SEM). **(G)** Representative immunofluorescence staining images of claudin-3, claudin-7, and EpCAM (*green*), and tubulin (*red*), in addition to secondary antibody controls, counterstained with DAPI in *blue* in two control and two *Tmprss2* knockout mCCD$_{cl1}$ cell colonies. Scale bar: 50 µm. Data were analysed by an unpaired *t* test (A, C, D, E). *P*-values < 0.05 were considered as statistically significant; ***P* < 0.001 and ****P* < 0.0001.
Source data are available for this figure.

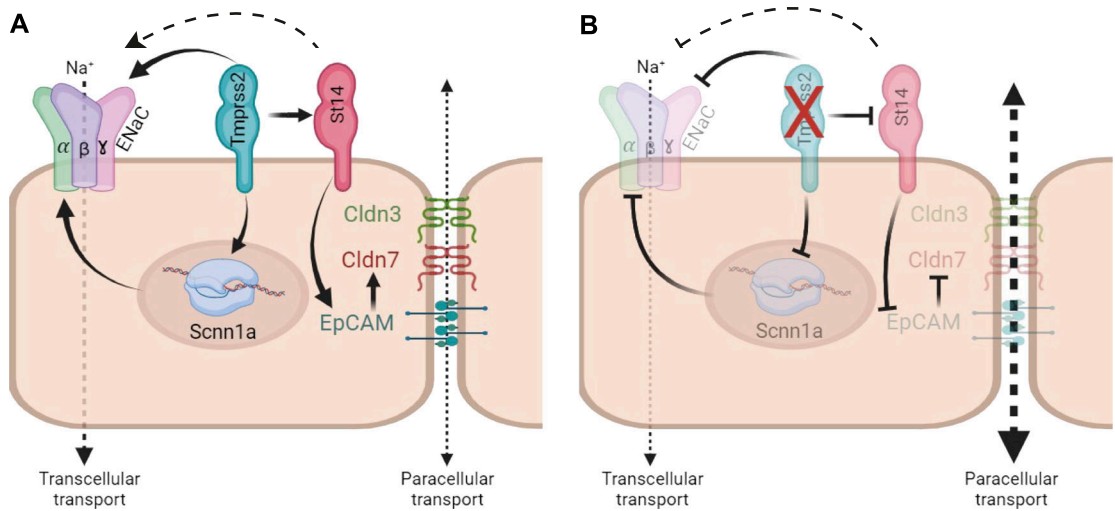

**Figure 9. Working model for the dual role of Tmprss2 in αENaC mRNA and protein abundance and EpCAM/claudin-7 complex and tight junction regulation.**
**(A)** Epithelial barrier integrity is maintained through proteases, ion channels, and tight junctions. **(B)** Tmprss2 deficiency induced a loss of the EpCAM/claudin-7 tight junction complex via down-regulation of CAP3/St14 and drastically reduced ENaC expression and function abolishing transepithelial sodium and paracellular transport in kidney cells; the tight epithelium becomes leaky.

**Table 1. Guide RNAs targeting Tmprss2 and controls targeting luciferase.**

| Name | Oligo sequence |
|---|---|
| *Tmprss2* sgRNA-1F | aaacCTGGTGGTCTCGGAGGACAGc |
| *Tmprss2* sgRNA-1R | caccgCTGTCCTCCGAGACCACCAG |
| *Tmprss2* sgRNA-2F | caccgTAGCAAGGTCCGATTCCTGG |
| *Tmprss2* sgRNA-2R | aaacCCAGGAATCGGACCTTGCTAc |
| Luciferase control sgRNA-1F | caccgACAACTTTACCGACCGCGCC |
| Luciferase control sgRNA-1R | aaacGGCGCGGTCGGTAAAGTTGTc |

Guide sequences in uppercase and lowercase letters are flanking overhang sequences.

Single-cell clones were subsequently expanded into colonies, and genomic DNA was extracted for genotyping by PCR using DNA primers F-5′-GCTGGCTCTCTCCCTGTTTT-3′ and R-5′-GCTCCTGAG-GACTTGGGATG-3′ and verified by sequencing with the forward primer. In addition to DNA sequencing, RNA was also extracted, and cDNA was synthesized, amplified using F-5′-GGCATTGAACT-CAGGGTCAC-3′ and R-5′-ATGGGTAGTACTGGGCTGGA-3′, and sequenced using the forward primer. To ensure genetic disruption because of Cas9, primers overlapping the cleavage sites were designed: F-5′-TCAGGGTCACCTCCAGGAATC-3′ and R-5′-CCATTGG-GAGCCACTGGTG-3′, and to precisely determine whether both regions were successful, the overlapping primers were used in combination with non-overlapping primers F-5′-TATGA-GAACCACGGGTATCAGT-3′ and R-5′-CGTTGTAATCCTCGGAGCATACT-3′ as illustrated in Fig S4B.

### Scnn1a-inducible overexpression system

A doxycycline-inducible Scnn1a mCCD<sub>cl1</sub> cell line was generated following the Lenti-X Tet-One Inducible Expression System User Manual (Takara Bio). The *Scnn1a* sequence was isolated from mCCD<sub>cl1</sub> cells using the forward 5′-CCCTCGTAAAGAATTCATGATGCTGGACCA-CACCAGAGCC-3′ primer containing an EcoRI restriction site and reverse 5′-ATCCGCCGGCACCGGTTCAGAGTGCCATGGCCGGAGCACA-3′ primer containing an AgeI restriction site. In brief, the *Scnn1a* gene was cloned into the pLVX-TetOne-Puro plasmid and sequenced using the 5′-GGATTAGGCAGTAGCTCTGACGGCCC-3′ primer to ensure correct insertion. The plasmid was co-transfected with the Lenti-X Packaging Shots into the Lenti-X 293T cell line, and lentiviral particles were harvested. mCCD<sub>cl1</sub> cells were then transduced with the viral particles, and selection was maintained with 2 µg/ml puromycin. For experimental analysis, the cells were treated with either 100 ng/ml or 1 µg/ml doxycycline (or $H_2O$ vehicle control) for 24 h before harvesting RNA and protein for ENaC current analysis.

### Real-time PCR analyses

mRNA was isolated using QIAGEN RNeasy Mini Kit according to the manufacturer's instructions, and reverse transcription and RT–PCR analyses from kidneys and cell lysates were performed as previously described (Keppner et al, 2019). Each measurement was performed in triplicate, and relative abundance of mRNA was calculated by normalization to *Gapdh* using the 2-ΔΔCt method. Real-time PCR primer sequences are listed in Table 2.

### Western blot analyses

Cultured mCCD<sub>cl1</sub> cells were directly lysed with radio-immunoprecipitation assay buffer, whereas organ samples were lysed with radioimmunoprecipitation assay and mechanical homogenization using a tissue lyser. Proteins were quantified by BCA assay, and equal amounts of protein were separated by SDS–PAGE, then transferred to nitrocellulose membranes. After protein visualization by Ponceau S dye and blocking with TBS/Tween buffer containing 5% milk, membranes were incubated at 4°C overnight

**Table 2. Primer sequences used for real-time PCR.**

| Name | Forward sequence 5'-3' | Reverse sequence 5'-3' |
|---|---|---|
| Tmprss2 | TGAATGTGAGCTCAGGCAAC | AGCGCAAAGAAACCACCATG |
| Scnn1a | GCACAACCGCATGAAGACG | AAAGCAAACTGCCAGTACATC |
| Scnn1g | CCGAGATCGAGACAGCAATGT | CGCTCAGCTTGAAGGATTCTG |
| Furin | TTGGCAGCTGGTATCATTGC | TAGCCCAATCATCAGCGTTG |
| Slc26a4 | TGTTGGCTGCATCCTTTTCC | ACGTTGCTTATCCCAAAGGC |
| Epcam | TTGCTCCAAACTGGCGTCTA | ACGTGATCTCCGTGTCCTTGT |
| Cldn2 | TGCGACACACAGCACAGGCATCAC | TCAGGAACCAGCGGCGAGTAGAA |
| Cldn3 | GCACCCACCAAGATCCTCTA | TCGTCTGTCACCATCTGGAA |
| Cldn7 | AAGCGAAGAAGGCCCGAATA | GCAAGACCTGCCACAATGAA |
| Gapdh | AGGTCGGTGTGAACGGATTTG | TGTAGACCATGTAGTTGAGGTCA |
| Snai1 | TGTCTGCACGACCTGTGGAAAG | CTTCACATCCGAGTGGGTTTGG |
| Snai3 | CACATTAGAACTCACACTGGGGA | TGCCCTCAGGTTTGATCTGTC |
| Twist 1 | GATTCAGACCCTCAAACTGGCG | AGACGGAGAAGGCGTAGCTGAG |
| Twist 2 | CAGCAAGATCCAGACGCTCAAG | ACACGGAGAAGGCGTAGCTGAG |
| Tjp1 | GTTGGTACGGTGCCCTGAAAGA | GCTGACAGGTAGGACAGACGAT |
| Ocln | TGGCAAGCGATCATACCCAGAG | CTGCCTGAAGTCATCCACACTC |
| Cdh1 | GGTCATCAGTGTGCTCACCTCT | GCTGTTGTGCTCAAGCCTTCAC |
| Prss8 | TGACCATTCTGCTCCTTCTC | GACACCACCCATTTATTTGACAC |
| St14 | TCCCTACCACAAGAAGTCGG | GCCACCACAGATGTTAGCAC |
| Hpn | ACATTGCTTTCCAGAGCGGA | AGAGGTGGACCAAGGCAATG |

**Table 3. Antibodies used for Western blotting and immunocytochemistry (ICC).**

| Antibody | Species | Western blot dilution | ICC dilution | Supplier |
|---|---|---|---|---|
| αENaC | Rabbit | 1:4,000 | | Kindly provided by J Loffing (Sorensen et al, 2013) |
| γENaC | Rabbit | 1:1,000 | | #SPC-405D; StressMarq |
| Furin | Rabbit | 1:2,000 | | #ab3467; Abcam |
| EpCAM | Rabbit | 1:1,000 | 1:500 | #PA5-19832; Invitrogen |
| Claudin-2 | Rabbit | 1:250 | | #51-6100; Invitrogen |
| Claudin-3 | Rabbit | 1:1,000 | 1:100 | #34-1700; Invitrogen |
| Claudin-7 | Rabbit | 1:250 | 1:125 | #34-9100; Invitrogen |
| Tubulin | Mouse | 1:2,000 | 1:1,000 | #T5168; Sigma-Aldrich |
| St14 | Rabbit | 1:500 | | #AF3946; R&D Systems |
| Prss8 | Sheep | 1:500 | | #15527-1-AP; ProteinTech |
| ZO-1 | Rabbit | 1:500 | | #61-7300; Zymed |
| Occludin | Rabbit | 1:500 | | #71-1500; Covance |
| E-cadherin | Mouse | 1:500 | | #610181; BD Biosciences |

with primary antibodies (Table 3), then 1 h with DyLight anti-mouse and anti-rabbit secondary antibodies (#35519 and #SA5-10036, 1:5,000; Invitrogen). Blots were visualized using the Odyssey classic scanner (LI-COR Biosciences), quantified using Image Studio Lite software, and normalized to tubulin expression. For Prss8, St14, ZO-1, occludin, and E-cadherin proteins, secondary anti-rabbit, anti-mouse, and anti-sheep antibodies (1:10,000; Amersham Biosciences) were detected by chemiluminescence (SuperSignal West Pico; Thermo Fisher Scientific).

### Immunocytochemistry

500,000 $mCCD_{cl1}$ cells were seeded onto glass coverslips and after 48 h fixed with 4% paraformaldehyde for 10 min. After washing with

**Life Science Alliance**

PBS, cells were blocked and permeabilized with PBS containing 0.3% Triton X and 3% BSA for 30 min, followed by incubation of primary antibodies in the same buffer for 1 h. The cells were then washed with PBS and incubated with Alexa Fluor 488 anti-rabbit and 647 anti-mouse secondary antibodies (#A-21206 and #A-21449; Invitrogen) for a further hour in PBS. Finally, coverslips were mounted onto slides using Fluoromount-G with DAPI mounting medium (#00-4959-52; Invitrogen). Images were acquired as a z-stack using an Olympus confocal microscope with a 63x oil objective, and all microscope parameters were kept constant from sample to sample.

## Mice

10- to 17-wk-old C57BL/6J mice (Janvier) were kept under standard conditions (0.17% sodium, 0.97% K$^+$) with food and water ad libitum. Organs were isolated and proceeded for further analyses through RT–PCR (snap-frozen in liquid N2). Animal maintenance and experimental procedures were approved by the local committee for animal experimentation (Service de la Consommation et des Affaires Vétérinaires, Lausanne, Vaud, Switzerland) (#VD3812a).

## RNAscope analyses

Kidneys from three male and three female C57BL/6J mice fed a standard diet were fixed in 10% formalin for 24 h at RT, and kidneys were cut transversally and longitudinally into 4- to 5-$\mu m$ slices. The RNAscope Multiplex Fluorescent V2 assay was performed according to the manufacturer's protocol on 4-$\mu m$ paraffin sections and hybridized with probes purchased from ACD Bio-Techne: Mm-*Scnn1a* (#441391-C3); Mm-*Tmprss2* (#496721); Mm-*Furin* (#864041); Mm-3-plex positive control *Polr2a-C1*, *Ppib-C2*, and *Ubc-C3* (#320881); and 3-plex negative control bacterial *DapB* (#320871); and revealed with TSA Opal570 (#FP1488001KT) or TSA Opal650 (#FP1496001KT). Tissues were counterstained with DAPI and mounted with Prolong Diamond Antifade Mountant (#P36965; Thermo Fisher Scientific). Whole kidney sections were imaged at 20x magnification using an AxioScan 7 slide scanner and images analysed in QuPath. For quantification, ROIs were drawn in regions of the cortex and medulla, cells were detected using the software's cell detection algorithm, then a signal measurement classifier was set for each channel, and then the number of single-positively, double-positively, or double-negatively stained cells was determined. For intensity measurements, the same QuPath parameters were used to quantify the mean cell signal intensity for each staining in the single-positive, double-positive, and double-negative cell populations. To normalize signal intensity, the background signal detected in the double-negative cell population was subtracted from the signal in the single- and double-positive cell population. For each mouse, one kidney was cut in a transversal plane, and the other in a sagittal plane; therefore, six male and six female kidneys were analysed. On average, ~17,000 cells were analysed in the cortex and ~9,000 cells in the medulla per transversal section, and ~22,000 cells in the cortex and ~13,000 cells in the medulla of sagittal sections.

## RNA sequencing

mRNA was harvested from three sequential passages, in triplicate, from two single control mCCD$_{cl1}$ cell clones and two *Tmprss2* knockout clones generated by CRISPR/Cas9 as described in the Methods section "CRISPR/Cas9 gene editing." RNA was extracted using RNeasy Mini Kit (74106; QIAGEN) according to the manufacturer's instructions. RNA quality was determined by fragment analyser, and all samples were considered good quality with an RNA quality number > 8.3. Library preparation, quality control, data processing, and statistical analysis were carried out by the Lausanne Genomics Technologies Facility. TruSeq mRNA libraries were loaded on an Illumina NovaSeq 6000 instrument and sequenced as single-end 100-bp reads.

## Data processing

Purity-filtered reads were adapted and quality-trimmed with Cutadapt (v. 2.5; Martin, 2011). Reads matching to ribosomal RNA sequences were removed with fastq_screen (v. 0.11.1). Remaining reads were further filtered for low complexity with reaper (v. 15-065; Davis et al, 2013). Reads were aligned against Mus_musculus.GRCm38 genome using STAR (v. 2.5.3a; Dobin et al, 2013). The number of read counts per gene locus was summarized with htseq-count (v. 0.9.1; Anders et al, 2015) using gene annotation. The quality of the RNA-seq data alignment was assessed using RSeQC (v. 2.3.7; Wang et al, 2012). The htseq-generated counts data were used for the analysis.

## Data transformation and quality control

Statistical analysis was performed in R (version 4.2.2). Genes with low counts were filtered out according to the rule of 1 count per million (cpm) in at least 1 sample. Library sizes were scaled using TMM normalization. Subsequently, the normalized counts were transformed to cpm values, and a log$_2$ transformation was applied by means of the function cpm with the parameter setting prior.counts = 1 (edgeR, v 3.30.3; Robinson et al, 2010). After data normalization, a quality control analysis was performed through sample correlation, clustering, and PCA, which revealed a possible batch effect because of harvesting and passage, and batch correction was performed using the "RemoveBatchEffect" function in limma (Ritchie et al, 2015).

## Statistical analysis with limma

Genes with low counts were filtered out according to the following rule: log$_2$ of normalized CPM > 1 in at least one sample. Differential expression was computed with the R/Bioconductor package limma (Ritchie et al, 2015) by fitting data to a linear model. Each of the four contrasts was first summarized separately using moderated t-statistics via the topTable limma function. *P*-values were adjusted using the Benjamini–Hochberg (BH) method, which controls for the FDR. After testing the DE genes for a FDR < 0.05, the genes were also filtered according to their fold change using the treat limma function. The log fold change cut-off was set to 1 so that only genes with an absolute fold change superior to 2 were kept.

## Data Availability

All data generated and analysed in this study are available from the corresponding author on reasonable request. The RNA-seq data from this publication have been deposited in the GEO database (https://www.ncbi.nlm.nih.gov/geo/; accession number GSE247775).

## Supplementary Information

## Acknowledgements

We like to thank all members of the Hummler laboratory for useful discussions. We like to acknowledge the excellent graphical work by Mia Braunwalder. This work was supported by the Swiss National Foundation, Grant FNRS 31003A-182478/1, and the National Center of Competence in Research "Kidney.CH," Lausanne, Switzerland (NCCR, N-403-07-23 to E Hummler). We thank the Histology Core Facility of the Ecole Polytechnique Fédérale de Lausanne (EPFL) for their technical input of the RNAscope experiments, the Lausanne Genomics Technologies Facility (GTF) from the University of Lausanne for the RNA-sequencing analysis, and the Cellular Imaging Facility (CIF) for the microscope use and guidance.

### Author Contributions

OJ Rickman: conceptualization, data curation, software, formal analysis, supervision, validation, investigation, visualization, methodology, and writing—original draft, review, and editing.
E Guignard: data curation, software, formal analysis, investigation, visualization, and methodology.
T Chabanon: data curation, formal analysis, investigation, and methodology.
G Bertoldi: data curation, formal analysis, investigation, and methodology.
M Auberson: data curation, formal analysis, validation, investigation, and writing—review and editing.
E Hummler: conceptualization, resources, data curation, formal analysis, supervision, funding acquisition, validation, investigation, visualization, methodology, project administration, and writing—original draft, review, and editing.

### Conflict of Interest Statement

The authors declare that they have no conflict of interest.

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
