## [Reviewer comments · Life Science Alliance]

Life Science Alliance

Tmprss2 maintains epithelial barrier integrity and transepithelial sodium transport

Olivia Rickman, Emma Guignard, Thomas Chabanon, Giovanni Bertoldi, Muriel Auberson, and Edith Hummler
DOI: <https://doi.org/10.26508/lsa.202302304>

Corresponding author(s): *Edith Hummler, University of Lausanne and Olivia Rickman, University of Lausanne*

Review Timeline:

Submission Date:	2023-08-04
Editorial Decision:	2023-09-06
Revision Received:	2023-11-29
Editorial Decision:	2023-12-19
Revision Received:	2023-12-21
Accepted:	2023-12-22

Transaction Report:

September 6, 2023

Re: Life Science Alliance manuscript #LSA-2023-02304-T

Dr. Edith Hummler
University of Lausanne
Departement of Biomedical Sciences
rue du Bugnon 27
Lausanne, Vaud CH-1011
Switzerland

Dear Dr. Hummler,

Thank you for submitting your manuscript entitled "Tmprss2 maintains epithelial barrier integrity and transepithelial sodium transport" to Life Science Alliance. The manuscript was assessed by expert reviewers, whose comments are appended to this letter. We invite you to submit a revised manuscript addressing the Reviewer comments.

Thank you for this interesting contribution to Life Science Alliance. We are looking forward to receiving your revised manuscript.

Sincerely,

B. MANUSCRIPT ORGANIZATION AND FORMATTING:

Reviewer #1 (Comments to the Authors (Required)):

In this manuscript, Rickman et al. examined the role of *tmprss2* in the regulation of tight junction barriers and sodium transport in the epithelium. They showed that the expression of *tmprss2* and *scnn1a*, a component of ENaC, overlap primarily in the kidney. Stimulation and inhibition of ENaC by aldosterone and amiloride, respectively, resulted in upregulation of *scnn1a* and *tmprss2* transcription. Exogenous expression of *scnn1a* also increased *tmprss2* transcription without affecting sodium current. The authors further analyzed *tmprss2* function using knockout cell clones. *Tmprss2* knockout cells showed a loss of barrier function as well as sodium current. RNA-seq analysis revealed reduced expression of tight junction-related genes in KO cells, which was inferred to be the cause of the loss of barrier function. The finding that the serine protease *tmprss2* regulates barrier function is interesting and will be of interest in the research field, but its regulatory mechanism is largely unknown and needs to be elucidated.

Main points

1. How does *tmprss2* regulate transcription of claudins and *epcam*. This needs to be clarified experimentally. Additionally, are transcription factors such as *snail*, *slug*, and *twist* known to be involved? Is the expression of adherens junction genes such as *e-cadherin* also decreased in KO cells? Is ENaC involved in the transcriptional regulation of tight junction genes by *tmprss2*?

2. ENaC is regulated by multiple mechanisms, including transcription, translation, localization, phosphorylation, proteolytic activation, and degradation. The authors need to distinguish the effects on ENaC in each experiment. e.g. *Scnn1a* overexpression induced upregulation of *tmprss2* but did not alter sodium current, suggesting that aldosterone-induced upregulation of *scnn1a* and *tmprss2* does not play a major role in changing sodium current. But the mechanism is not investigated.

Minor points.

3. page 5, line 5, "*Scnn1a* is highly dependent on *Tmprss2* expression" is incorrect. The function of *tmprss2* and whether its expression affects *scnn1a* has not yet been tested here.

4. figure 8. Does exogenous expression of *tmprss2* in KO cells reverse the effect of knockout?

5. In the first paragraph of the Discussion, the authors argue that "Our data clearly indicate that *Tmprss2* is closely linked to the upregulation of *Scnn1a*. These findings are in agreement with two former studies that reported a 2.6- and 3-fold increase in ENaC-mediated sodium current when *Tmprss2* was co-injected with ENaC-subunits into the *Xenopus* oocytes.". Since the authors' findings relate to transcriptional regulation of the ENaC alpha subunit and the other studies relate to proteolytic regulation of the ENaC gamma subunit, it is difficult to correlate them as "in agreement".

6. fig. 2 It would be helpful if the authors could clarify the type of tubules that the DP cells belong to experimentally (probably collecting ducts?).

Reviewer #2 (Comments to the Authors (Required)):

In this manuscript Rickman and co-workers investigated the role of the transmembrane serine protease *TMPRSS2* in regulation and maintenance of the epithelial barrier integrity and epithelial sodium channel (ENaC) activation in murine kidney cells. The authors show that *TMPRSS2* is co-expressed with the alpha subunit of ENaC (*Scnna1*) in murine kidney cells. Furthermore, the authors show that knockout of *TMPRSS2* downregulates expression of *Scnna1* as well as tight junction proteins *EpCAM*, *claudin3* and *claudin7*.

The manuscript is well written and the data are very interesting and extend previous knowledge on a possible role of *TMPRSS2* in the activation of the ENaC. The study is well suited for publication in Life Science Alliance, however, a few major points still need to be clarified before the work can be published.

The main criticism is that only *TMPRSS2* was investigated. A study by Higashi et al., 2023 et al., which is also referred to in the

manuscript, recently presented a working model of tight junction maintenance and repair via a EpCAM-claudin7 complex and release of claudin7 due to proteolytic cleavage of this complex. Importantly, they showed that different transmembrane serine proteases (hepsin, prostasin, TMPRSS14, TMPRSS4) are involved in cleavage of the EpCAM-claudin7 complex in the canine kidney cell line MDCK. Only knockout of all 4 proteases resulted in impaired barrier function. In the present work, Rickman and co-workers only focused on the role and (mRNA) expression of TMPRSS2 in ENaC activation and maintenance of the tight junction barrier in murine kidney cells. The authors should extend the investigations to other proteases (such as hepsin, prostasin, TMPRSS4, TMPRSS14), e.g. include these proteases in the studies shown in Figures 2, 3 and 4. I would expect that knockout of other proteases would reduce Scann1 expression and activity even more dramatically.

Also arguing against a sole major role of TMPRSS2 in the regulation of the epithelial barrier in mice is that TMPRSS2-KO mice do not show an obvious phenotype. If TMPRSS2 is critical for regulation of ENaC function and maintenance of tight junction barrier in the kidney, as discussed by the authors, TMPRSS2-deficient mice should show abnormalities. The authors should discuss this and, if necessary, tone down their conclusions. ("...our data identify Tmprss2 as critical protease in the maintenance of the tight junction barrier and regulation of ENaC.")

Figure 3A-D) The mRNA and protein expression levels of Scnn1a and furin do not always correlate (for unknown reasons). For TMPRSS2, the authors examined only mRNA expression. However, due to the differences in mRNA versus protein expression observed for Scann1 and Furin upon aldosterone or amilorid treatment, protein expression should also be determined for TMPRSS2 to draw definitive conclusions about altered expression.

Reviewer #3 (Comments to the Authors (Required)):

This study shows for distal kidney tubular epithelia (mice and murine cell monolayers) that lack of Tmprss2 impairs both, the paracellular barrier and ENaC function. As a mediator, Scnn1a has shown to be involved. The results and conclusions of this manuscript provide an advance in the field of epithelial transport and barrier research, as it explains the regulation and function of a transcellular transport pathway (ENaC) with a paracellular pathway (tight junction proteins).

Most of the specific issues are minor:

(1) The Abstract should be improved regarding didactic clarity. The graphical abstract is better in this respect and may serve as a red thread.

(2) p 3, para 3: "In this study ..." This sentence is somehow incomplete and should be re-worded.

(3) P 3, para 3: "results in an impaired paracellular transport"  "results in impaired transepithelial transport and paracellular barrier"

(4) p 4, para 3: "we applied aldosterone in increasing concentrations" and "Aldosterone significantly induced a dose-dependent increase". As only two concentrations were used, "dose dependent" is overdone. More importantly, both concentrations are non-physiologically high. 300 nM is clearly in the glucocorticoid range and thus may have additional effects. This limitation has at least to be discussed.

(5) p 4, para 5 and elsewhere: Different terms like "transepithelial sodium current", "transepithelial ENaC current", "amiloride-sensitive ENaC current", "ENaC-mediated sodium current", "ENaC current", and "ENaC short circuit currents" are used throughout the text, although always the same is meant. To avoid confusion, only one term should be used for one identical parameter. How about "ENaC current", after having explained somewhere that amiloride at a concentration of 10 μ M selectively but fully blocks ENaC-mediated transepithelial sodium current.

(6) p 10, para 2: The area (cm^2) and the pore diameter (μm) of the filters should be provided.

(7) p 10, para 2: "10 μ M amiloride transepithelial voltage and resistance measurements were made ..." This is misleading because it suggests that the difference before and after adding amiloride is measured (amiloride-sensitive current). However, in Figs. 3F, 4D, 5D-E-F, S3A "plain" values were given and amiloride was added only at the end of the experiment.

(8) p 10, para 2: Are the electrical measurements performed under temperature control (37 C)?

(9) Fig 5D: Resistance should be given with reference to the tissue area  $\Omega\cdot\text{cm}^2$

(10) Graphical abstract: " Tight epithelia" " Leaky epithelia" may suggest that the two drawings represent tight epithelia and leaky epithelia in general. The right drawing does not represent "leaky epithelia" but the former tight epithelium having turned leaky after Tmprss2 KO.

Suggestion: "Tight epithelium" "... becoming leaky".

The dashed lines indicating paracellular transport should have arrow heads on both ends.

Several occasions throughout the text:

(11) Hyphens should be added where appropriate, e.g. amiloride-sensitive, ENaC-activating, collagen-treated, and according.

(12) Generally, a space should be added between number and dimension (e.g. 1 μ M  1 μ M).

(13) "significant decrease", "significantly downregulates", "significantly higher" and according wording: If a change is reported, delete "significant" because this is self-understanding -- otherwise it could not be claimed different. Of course, if data are not statistically different, then it reads "not significantly different".

Rickman et al., #LSA-2023-02304-T

Responses to Referees

We would like to thank the reviewers for their positive and constructive comments and suggestions. We now included 2 new panels of figures and made following modifications:

- New data was added illustrating mRNA and protein expression of CAP3/St14 (modified Fig. 8) and of CAP1/Prss8 (prostasin), ZO-1 (zona occludens-1), occludin and E-cadherin in *Tmprss2* control and knockout cells as requested (new Fig. S5)
- A new Fig. 9 (modified graphical abstract) was added as proposed as “working model”.
- The abstract was rewritten and text was amended as recommended by the reviewers.

Reviewer #1 (Comments to the Authors (Required)):

In this manuscript, Rickman et al. examined the role of *tmprss2* in the regulation of tight junction barriers and sodium transport in the epithelium. They showed that the expression of *tmprss2* and *scnn1a*, a component of ENaC, overlap primarily in the kidney. Stimulation and inhibition of ENaC by aldosterone and amiloride, respectively, resulted in upregulation of *scnn1a* and *tmprss2* transcription. Exogenous expression of *scnn1a* also increased *tmprss2* transcription without affecting sodium current. The authors further analyzed *tmprss2* function using knockout cell clones. *Tmprss2* knockout cells showed a loss of barrier function as well as sodium current. RNA-seq analysis revealed reduced expression of tight junction-related genes in KO cells, which was inferred to be the cause of the loss of barrier function. The finding that the serine protease *tmprss2* regulates barrier function is interesting and will be of interest in the research field, but its regulatory mechanism is largely unknown and needs to be elucidated.

Main points

1. How does *tmprss2* regulate transcription of claudins and *epcam*. This needs to be clarified experimentally.

Tmprss2 is primarily known for its role in proteolytic cleavage of proteins like the angiotensin converting enzyme 2 (ACE2) (Wettstein et al., 2022) or the protease-activated receptor 2 (PAR2) involved in various signaling pathways rather than direct transcriptional regulation of claudins and EpCAM (Wilson et al., 2005). A further endogenous substrate identified is the epithelial sodium channel (Donaldson et al., 2002)(Sure et al., 2022). The physiological role of *Tmprss2* however is still largely unknown. *Tmprss2* may indirectly affect the expression of genes by cleaving and activating certain signaling molecules or receptors which in turn can modulate downstream gene expression. This is now specified in the manuscript. We feel however that it is out of the scope of this paper and should be addressed in follow up studies.

Additionally, are transcription factors such as *snail*, *slug*, and *twist* known to be involved? The transcription factors *Snail1* and *Twist2* are absent while *Slug* and *Twist1* are very low abundantly expressed in our mCCD RNA-seq data as well as in the RNA-seq data from Chen et al., 2021 that comprises a comprehensive map of mRNAs and their isoforms along the mouse renal tubule (Chen et al., 2021). We now performed qPCRs of these transcription factors in *Tmprss2* control and knockout cells and confirmed the absence or

low expression of these transcription factors. The mRNA expression of the transcription factors like eg., slug or twist1 implicated in trans-suppression of EpCAM by binding to the E-box elements in the promoter (Liu et al., 2021) seemed not to be involved. Those were low abundantly expressed in renal cells and not altered in Tmprss2 control versus knockout cells (data not shown). These findings are now discussed in the manuscript.

Is the expression of adherens junction genes such as e-cadherin also decreased in KO cells?

Our mCCD RNA-seq data revealed no change between in E-cadherin (*Cdh1*) expression control and knockout cells which is also confirmed by new qPCR analyses. This is also confirmed on the protein level (**new Figure S5**). Additionally, we analysed the mRNA and protein expression of the tight junction proteins ZO-1(*Tjp1*) and occludin (*Ocln*) which did not differ between Tmprss2 control and knockout cells. This is now included as **new Figure S6** and discussed.

Is ENaC involved in the transcriptional regulation of tight junction genes by tmprss2?

As shown in Figure S6 of our manuscript, we found no evidence that mRNA as well as protein abundances of EpCAM, claudin-3 or claudin-7 are altered when α ENaC abundance is increased either by aldosterone or in an inducible system. Tmprss2 is not known to be a direct transcriptional regulator of tight junctions. Although several proteases were proposed to be included in the EpCAM proteolysis (Higashi et al., 2023), only CAP3/St14 (matriptase) was significantly reduced in our Tmprss2 ko RNA-seq data. We newly validated downregulation of CAP3/St14 (matriptase) on the mRNA and protein expression level (**modified Figure 8**) and demonstrate that transcription and translation of CAP1/Prss8 (prostasin) is not affected (**new Figure S5**). CAP2/Tmprss4 and Tmprss1 (hepsin) are not detected in our RNA-seq data which is confirmed in the mouse renal tubule RNA-seq data performed by Chen et al. 2021 (Chen et al., 2021). These data are now newly included into the manuscript as **new Fig. S5** and **modified Fig.8** and discussed.

2. EnaC is regulated by multiple mechanisms, including transcription, translation, localization, phosphorylation, proteolytic activation, and degradation. The authors need to distinguish the effects on ENaC in each experiment. e.g. Scnn1a overexpression induced upregulation of tmprss2 but did not alter sodium current, suggesting that aldosterone-induced upregulation of scnn1a and tmprss2 does not play a major role in changing sodium current. But the mechanism is not investigated.

The aldosterone treatment of mCCD cells resulted in an increase of sodium current, an increased α ENaC mRNA and protein expression and an increase in Tmprss2 mRNA expression. Blocking ENaC function by amiloride resulted in a decrease of α ENaC protein and ENaC-mediated sodium current, but upregulation of α ENaC mRNA expression level that we interpret to compensation for the lacking ENaC function (**Fig. 3**). On the other side, the serine protease inhibitors aprotinin and camostat mesylate partly blocked ENaC-mediated current (**Figure S3**), indicating that serine proteases are not fully blocked by these inhibitors. This has been also previously reported (Sure et al., 2022)(Andreasen et al., 2006)). The regulation of ENaC is indeed quite complex and still not completely understood. We thus agree that with the experiments performed, we

cannot state that dysregulation of the epithelial barrier integrity may lead to dysfunction of transepithelial sodium transport or vice versa. This is now better explained.

Minor points.

3. page 5, line 5, "Scnn1a is highly dependent on Tmprss2 expression" is incorrect. The function of tmprss2 and whether its expression affects scnn1a has not yet been tested here.

We agree with the reviewer and modified the text accordingly. This statement is now replaced by "Following Tmprss2 knockout, aldosterone treatment and Scnn1a (α ENaC) overexpression, and amiloride block of ENaC, α ENaC mRNA and protein expression followed the Tmprss2 mRNA expression."

4. figure 8. Does exogenous expression of tmprss2 in KO cells reverse the effect of knockout?

We planned and performed this experiment, but unfortunately, we could not stably overexpress Tmprss2 in mCCD cells. This might be due to incompatibility with survival as it has been equally not successful with other membrane-bound serine proteases.

5. In the first paragraph of the Discussion, the authors argue that "Our data clearly indicate that Tmprss2 is closely linked to the upregulation of Scnn1a. These findings are in agreement with two former studies that reported a 2.6- and 3-fold increase in ENaC-mediated sodium current when Tmprss2 was co-injected with ENaC-subunits into the *Xenopus* oocytes.". Since the authors' findings relate to transcriptional regulation of the ENaC alpha subunit and the other studies relate to proteolytic regulation of the ENaC gamma subunit, it is difficult to correlate them as "in agreement".

We agree with the reviewer; this is now corrected in the text.

6. fig. 2 It would be helpful if the authors could clarify the type of tubules that the DP cells belong to experimentally (probably collecting ducts?).

It is generally accepted in the literature that α ENaC (Scnn1a) mRNA expression is found primarily in the second part of distal cortical collection duct (DCT2), in the cortical connecting tubule (CNT) as well as along the cortical collecting duct (CCD). This is confirmed by the recent comprehensive map of mRNAs across all renal tubule segments performed by Chen and coworker, (Chen et al., 2021). Tmprss2 mRNA expression is low abundant in the proximal tubule and is rather detected from the DCT onwards, thus primarily overlapping the ENaC subunit expression. This is now specified within the manuscript.

Reviewer #2 (Comments to the Authors (Required)):

In this manuscript Rickman and co-workers investigated the role of the transmembrane serine protease TMPRSS2 in regulation and maintenance of the epithelial barrier integrity and epithelial sodium channel (ENaC) activation in murine kidney cells. The authors show that TMPRSS2 is co-expressed with the alpha subunit of ENaC (Scnna1) in murine kidney cells. Furthermore, the authors show that knockout of TMPRSS2 downregulates expression of Scnna1 as well as tight junction proteins EpCAM, claudin3 and claudin7.

The manuscript is well written and the data are very interesting and extend previous knowledge on a possible role of TMPRSS2 in the activation of the ENaC. The study is well suited for publication in Life Science Alliance, however, a few major points still need to be clarified before the work can be published.

We thank the reviewer for this positive comment.

The main criticism is that only TMPRSS2 was investigated. A study by Higashi et al., 2023 et al., which is also referred to in the manuscript, recently presented a working model of tight junction maintenance and repair via a EpCAM-claudin7 complex and release of claudin7 due to proteolytic cleavage of this complex. Importantly, they showed that different transmembrane serine proteases (hepsin, prostasin, TMPRSS14, TMPRSS4) are involved in cleavage of the EpCAM-claudin7 complex in the canine kidney cell line MDCK. Only knockout of all 4 proteases resulted in impaired barrier function. In the present work, Rickman and co-workers only focused on the role and (mRNA) expression of TMPRSS2 in ENaC activation and maintenance of the tight junction barrier in murine kidney cells. The authors should extend the investigations to other proteases (such as hepsin, prostasin, TMPRSS4, TMPRSS14), e.g. include these proteases in the studies shown in Figures 2, 3 and 4. I would expect that knockout of other proteases would reduce Scann1 expression and activity even more dramatically.

At the time we started the project of Tmprss2 as a ENaC channel-activating protease, a link of membrane-bound serine proteases (MASPs) to EpCAM/claudin-7 complex was not yet published.

Although several proteases were proposed to be included in the EpCAM proteolysis in MDCK II cells (Higashi et al., 2023), mCCDcl1 cells do not express CAP2/Tmprss4 and Tmprss1 (hepsin), and only express low abundantly CAP1/Prss8, but abundantly CAP3/St14 (Tmprss14) (see also new Fig. S5, modified Fig. 8D-E). Higashi and coworkers did not exclude the required activation by other MASPs, since the expression of single MASP was not sufficient to support the barrier (Higashi et al., 2023). Only CAP3/St14 (matriptase) was significantly reduced in our Tmprss2 ko RNA-seq data. We newly validated downregulation of CAP3/St14 (matriptase) on the mRNA and protein expression level (modified Fig.8) and demonstrate that transcription and translation of CAP1/Prss8 (prostasin) is not affected (new Figure S5). CAP2/Tmprss4 and Tmprss1 (hepsin) are neither detected in our RNA-seq data nor by qPCR which is confirmed in the mouse renal tubule RNA-seq data performed by Chen et al. 2021 (Chen et al., 2021).

Indeed, contrary to their paper where single or a combination of MASPs transiently induced epithelial barrier leaks, the knockout of Tmprss2 in our study induced a sustained abolishment of ENaC-mediated transepithelial sodium transport and near-complete loss of EpCAM/claudin-3 protein abundance in mCCD cells. Higashi and coworkers suggested that “the barrier function of MASP-qKO cells was restored through “other” proteinase activity on EpCAM”. Our data highly suggest that Tmprss2 acts upstream of one of the analysed MASPs, namely CAP3/St14, and induced a sustained loss of protein abundance in Tmprss2 ko cells.

It is worth mentioning that MDCK II cells do not express Tmprss2. We like also to refer to previous and recently published CAP2/Tmprss4, CAP1/Prss8 and CAP3/St14 (Tmprss14, matriptase) knockouts in the renal tubule. While in CAP3/St14 (Tmprss14) ko mice α , β and γ ENaC protein abundances were reduced but not abolished (Ehret et al., 2023), Fig.1, no difference was seen in CAP2/Tmprss4 (Keppner et al., 2015) and CAP1/Prss8 knockout mice (Ehret et al., 2022), Fig. 4.

This is now better discussed in our manuscript.

Also arguing against a sole major role of TMPRSS2 in the regulation of the epithelial barrier in mice is that TMPRSS2-KO mice do not show an obvious phenotype. If TMPRSS2 is critical for regulation of ENaC function and maintenance of tight junction barrier in the kidney, as discussed by the authors, TMPRSS2-deficient mice should show abnormalities. The authors should discuss this and, if necessary, tone down their conclusions. ("...our data identify Tmprss2 as critical protease in the maintenance of the tight junction barrier and regulation of ENaC.")

We thank the reviewer for this comment. The Tmprss2 knockout by Kim and coworkers (Kim et al., 2006) replaced exon 10-13 by a neomycin resistance gene targeting thus 2 of the 3 relevant amino acids of the catalytic triad. Indeed, the authors of the Tmprss2 knockout demonstrated that their knockout is not complete, and "only a 20% reduction in Tmprss2 transcripts was observed in Tmprss2^{-/-} mice compared to WT mice....". It is therefore likely that these mice although not tested in the paper only lack a catalytically active protease, but still make aberrant and likely functional transcripts. A catalytically inactive protease might still exert other functions independent of its catalytic domain and interact with other proteins. This had been demonstrated for the serine protease CAP1/Prss8 with mice viable without catalytic domain (Peters et al., 2014)). The kidney phenotype of Tmprss2 "knockout" mice was not yet analysed. The incomplete knockout explains the lack of phenotype. This is now included in the manuscript.

Figure 3A-D) The mRNA and protein expression levels of Scnn1a and furin do not always correlate (for unknown reasons). For TMPRSS2, the authors examined only mRNA expression. However, due to the differences in mRNA versus protein expression observed for Scnn1 and Furin upon aldosterone or amilorid treatment, protein expression should also be determined for TMPRSS2 to draw definitive conclusions about altered expression.

Indeed, we tested all Tmprss2 antibodies available on the market and even tried to generate our own Tmprss2 antibodies via a company, so far without any success. We agree with the reviewer that functional Tmprss2 antibodies would be informative.

Reviewer #3 (Comments to the Authors (Required)):

This study shows for distal kidney tubular epithelia (mice and murine cell monolayers) that lack of Tmprss2 impairs both, the paracellular barrier and ENaC function. As a mediator, Scnn1a has shown to be involved. The results and conclusions of this manuscript provide an advance in the field of epithelial transport and barrier research, as it explains the regulation and function of a transcellular transport pathway (ENaC) with a paracellular pathway (tight junction proteins).

We like to thank this reviewer for the positive comment.

Most of the specific issues are minor:

- (1) The Abstract should be improved regarding didactic clarity. The graphical abstract is better in this respect and may serve as a red thread.

The abstract is now rewritten for didactic clarity. We further integrated the graphical abstract as working model in the manuscript as **new Fig. 9**. Text is amended accordingly.

- (2) p 3, para 3: "In this study ..." This sentence is somehow incomplete and should be reworded.

This is now corrected.

- (3) P 3, para 3: "results in an impaired paracellular transport"  "results in impaired transepithelial transport and paracellular barrier"

This is now corrected.

- (4) p 4, para 3: "we applied aldosterone in increasing concentrations" and "Aldosterone significantly induced a dose-dependent increase". As only two concentrations were used, "dose dependent" is overdone. More importantly, both concentrations are non-physiologically high. 300 nM is clearly in the glucocorticoid range and thus may have additional effects. This limitation has as least to be discussed.

Indeed, the concentration of 300 μM is relatively high and out of the physiological range, but the aim was to maximally induce the αENaC mRNA and protein abundance. This is now better discussed.

- (5) p 4, para 5 and elsewhere: Different terms like "transepithelial sodium current", "transepithelial ENaC current", "amiloride-sensitive ENaC current", "ENaC-mediated sodium current", "ENaC current", and " ENaC short circuit currents" are used throughout the text, although always the same is meant. To avoid confusion, only one term should be used for one identical parameter. How about "ENaC current", after having explained somewhere that amiloride at a concentration of 10 μM selectively but fully blocks ENaC-mediated transepithelial sodium current.

This is now unified and corrected throughout the text.

- (6) p 10, para 2: The area (cm^2) and the pore diameter (μm) of the filters should be provided.

The information (4.7 cm^2 , pore: 0.4 μm (polycarbonate membrane) is now included into the Material & Methods section.

- (7) p 10, para 2: "10 μM amiloride transepithelial voltage and resistance measurements were made ..." This is misleading because it suggests that the difference before and after adding amiloride is measured (amiloride-sensitive current). However, in Figs. 3F, 4D, 5D-E-F, S3A "plain" values were given and amiloride was added only at the end of the experiment.

The text is now corrected in the Material & Methods section.

- (8) p 10, para 2: Are the electrical measurements performed under temperature control (37 $^{\circ}\text{C}$)?

The experiments were performed at RT, but we meanwhile performed experiments at 37 $^{\circ}\text{C}$ which gave the same result (data not shown). The temperature is now specified in the manuscript.

- (9) Fig 5D: Resistance should be given with reference to the tissue area  $\Omega\cdot\text{cm}^2$

This is now added in the Fig. 5.

(10) Graphical abstract: " Tight epithelia" " Leaky epithelia" may suggest that the two drawings represent tight epithelia and leaky epithelia in general. The right drawing does not represent "leaky epithelia" but the former tight epithelium having turned leaky after *Tmprss2* KO.
Suggestion: "Tight epithelium" "... becoming leaky".
As proposed we now integrated the graphical abstract as working model (new Fig. 9) into the text. The comment by the reviewer is respected.

The dashed lines indicating paracellular transport should have arrow heads on both ends.
This is now corrected.

Several occasions throughout the text:

(11) Hyphens should be added where appropriate, e.g. amiloride-sensitive, ENaC-activating, collagen-treated, and according.
This is now checked throughout the text and corrected.

(12) Generally, a space should be added between number and dimension (e.g. 1µM  1 µM).
This is now corrected.

(13) "significant decrease", "significantly downregulates", "significantly higher" and according wording: If a change is reported, delete "significant" because this is self-understanding -- otherwise it could not be claimed different. Of course, if data are not statistically different, then it reads "not significantly different".
We agree with the reviewer; and corrected it throughout the text.

References

- Andreasen, D., Vuagniaux, G., Fowler-Jaeger, N., Hummler, E., & Rossier, B. C. (2006). Activation of epithelial sodium channels by mouse channel activating proteases (mCAP) expressed in *Xenopus* oocytes requires catalytic activity of mCAP3 and mCAP2 but not mCAP1. *Journal of the American Society of Nephrology: JASN*, 17(4), 968-976. <https://doi.org/10.1681/ASN.2005060637>
- Chen, L., Chou, C.-L., & Knepper, M. A. (2021). A Comprehensive Map of mRNAs and Their Isoforms across All 14 Renal Tubule Segments of Mouse. *Journal of the American Society of Nephrology: JASN*, 32(4), 897-912. <https://doi.org/10.1681/ASN.2020101406>
- Donaldson, S. H., Hirsh, A., Li, D. C., Holloway, G., Chao, J., Boucher, R. C., & Gabriel, S. E. (2002). Regulation of the epithelial sodium channel by serine proteases in human airways. *The Journal of Biological Chemistry*, 277(10), 8338-8345. <https://doi.org/10.1074/jbc.M105044200>
- Ehret, E., Jäger, Y., Sergi, C., Méritat, A.-M., Peyrollaz, T., Anand, D., Wang, Q., Ino, F., Maillard, M., Kellenberger, S., Gautschi, I., Szabo, R., Bugge, T. H., Vogel, L. K., Hummler, E., & Frateschi, S. (2022). Kidney-Specific CAP1/Prss8-Deficient Mice Maintain ENaC-Mediated Sodium Balance through an Aldosterone Independent Pathway. *International Journal of Molecular Sciences*, 23(12), 6745. <https://doi.org/10.3390/ijms23126745>

- Ehret, E., Stroh, S., Auberson, M., Ino, F., Jäger, Y., Maillard, M., Szabo, R., Bugge, T. H., Frateschi, S., & Hummler, E. (2023). Kidney-Specific Membrane-Bound Serine Proteases CAP1/Prss8 and CAP3/St14 Affect ENaC Subunit Abundances but Not Its Activity. *Cells*, 12(19), 2342. <https://doi.org/10.3390/cells12192342>
- Higashi, T., Saito, A. C., Fukazawa, Y., Furuse, M., Higashi, A. Y., Ono, M., & Chiba, H. (2023). EpCAM proteolysis and release of complexed claudin-7 repair and maintain the tight junction barrier. *The Journal of Cell Biology*, 222(1), e202204079. <https://doi.org/10.1083/jcb.202204079>
- Keppner, A., Andreasen, D., Mérillat, A.-M., Bapst, J., Ansermet, C., Wang, Q., Maillard, M., Malsure, S., Nobile, A., & Hummler, E. (2015). Epithelial Sodium Channel-Mediated Sodium Transport Is Not Dependent on the Membrane-Bound Serine Protease CAP2/Tmprss4. *PloS One*, 10(8), e0135224. <https://doi.org/10.1371/journal.pone.0135224>
- Kim, T. S., Heinlein, C., Hackman, R. C., & Nelson, P. S. (2006). Phenotypic analysis of mice lacking the Tmprss2-encoded protease. *Molecular and Cellular Biology*, 26(3), 965-975. <https://doi.org/10.1128/MCB.26.3.965-975.2006>
- Liu, X., Feng, Q., Zhang, Y., Zheng, P., & Cui, N. (2021). Absence of EpCAM in cervical cancer cells is involved in slug-induced epithelial-mesenchymal transition. *Cancer Cell International*, 21(1), 163. <https://doi.org/10.1186/s12935-021-01858-3>
- Peters, D. E., Szabo, R., Friis, S., Shylo, N. A., Uzzun Sales, K., Holmbeck, K., & Bugge, T. H. (2014). The membrane-anchored serine protease prostaticin (CAP1/PRSS8) supports epidermal development and postnatal homeostasis independent of its enzymatic activity. *The Journal of Biological Chemistry*, 289(21), 14740-14749. <https://doi.org/10.1074/jbc.M113.541318>
- Sure, F., Bertog, M., Afonso, S., Diakov, A., Rinke, R., Madej, M. G., Wittmann, S., Gramberg, T., Korbmayer, C., & Ilyashin, A. V. (2022). Transmembrane serine protease 2 (TMPRSS2) proteolytically activates the epithelial sodium channel (ENaC) by cleaving the channel's γ -subunit. *Journal of Biological Chemistry*, 298(6), 102004. <https://doi.org/10.1016/j.jbc.2022.102004>
- Wettstein, L., Kirchhoff, F., & Münch, J. (2022). The Transmembrane Protease TMPRSS2 as a Therapeutic Target for COVID-19 Treatment. *International Journal of Molecular Sciences*, 23(3), 1351. <https://doi.org/10.3390/ijms23031351>
- Wilson, S., Greer, B., Hooper, J., Zijlstra, A., Walker, B., Quigley, J., & Hawthorne, S. (2005). The membrane-anchored serine protease, TMPRSS2, activates PAR-2 in prostate cancer cells. *The Biochemical Journal*, 388(Pt 3), 967-972. <https://doi.org/10.1042/BJ20041066>

December 19, 2023

RE: Life Science Alliance Manuscript #LSA-2023-02304-TR

Dr. Edith Hummler
University of Lausanne
Departement of Biomedical Sciences
rue du Bugnon 27
Lausanne, Vaud CH-1011
Switzerland

Dear Dr. Hummler,

Thank you for submitting your revised manuscript entitled "Tmprss2 maintains epithelial barrier integrity and transepithelial sodium transport". We would be happy to publish your paper in Life Science Alliance pending final revisions necessary to meet our formatting guidelines.

- please add ORCID ID for the secondary corresponding author--they should have received instructions on how to do so
- please add the Twitter handle of your host institute/organization as well as your own or/and one of the authors in our system
- Tables should be numbered consecutively with Arabic numerals (1, 2, 3, 4). They can be included at the bottom of the main manuscript file or sent as separate files.
- please remove label A from Figure S2 since it has only one panel
- please add callouts for Figures 1A-B; 8F; 9A-B; S1A-B; S5B-H; S6A-F to your main manuscript text
- you may consider uploading Figure 9 as a Graphical Abstract, rather than as a figure, but this is up to you

Figure Checks:

-it looks like there may be a splice in the blot in Figure S5B, just to the left of the WT column, however, this could just be how the gel ran. If the blot was indeed cut and spliced, please indicate this with a black vertical line and mention what the line represents in the legend.

A. FINAL FILES:

B. MANUSCRIPT ORGANIZATION AND FORMATTING:

Sincerely,

Reviewer #1 (Comments to the Authors (Required)):

Due to technical difficulties, some concerns remain unresolved, but the revised manuscript is much improved. This reviewer considers the manuscript acceptable.

Reviewer #2 (Comments to the Authors (Required)):

The authors have revised the manuscript very well and taken all comments into account. The manuscript can be published in this form.

December 22, 2023

RE: Life Science Alliance Manuscript #LSA-2023-02304-TRR

Dr. Edith Hummler
University of Lausanne
Departement of Biomedical Sciences
rue du Bugnon 27
Lausanne, Vaud CH-1011
Switzerland

Dear Dr. Hummler,

Thank you for submitting your Research Article entitled "Tmprss2 maintains epithelial barrier integrity and transepithelial sodium transport". It is a pleasure to let you know that your manuscript is now accepted for publication in Life Science Alliance. Congratulations on this interesting work.

DISTRIBUTION OF MATERIALS:

Again, congratulations on a very nice paper. I hope you found the review process to be constructive and are pleased with how the manuscript was handled editorially. We look forward to future exciting submissions from your lab.

Sincerely,
